# COUNTERFACTUAL RATIONALITY:
# A CAUSAL APPROACH TO GAME THEORY

## ABSTRACT

The tension between rational and irrational behaviors in human decision-making has been acknowledged across a wide range of disciplines, from philosophy to psychology, neuroscience to behavioral economics. Models of multi-agent interactions, such as von Neumann and Morgenstern's expected utility theory and Nash's game theory, provide rigorous mathematical frameworks for how agents should behave when rationality is sought. However, the rationality assumption has been extensively challenged, as human decision-making is often irrational, influenced by biases, emotions, and uncertainty, which may even have a positive effect in certain cases. Behavioral economics, for example, attempts to explain such irrational behaviors, including Kahneman's dual-process theory and Thaler's nudging concept, and accounts for deviations from rationality. In this paper, we analyze this tension through a causal lens and develop a framework that accounts for rational and irrational decision-making, which we term *Causal Game Theory*. We then introduce a novel notion called counterfactual rationality, which allows agents to make choices leveraging their irrational tendencies. We extend the notion of Nash Equilibrium to counterfactual actions and Pearl Causal Hierarchy (PCH), and show that strategies following counterfactual rationality dominate strategies based on standard game theory. We further develop an algorithm to learn such strategies when not all information about other agents is available.

## 1 INTRODUCTION

Decision-making in multi-agent systems (MAS) is a critical problem with broad applications across disciplines such as economics, social sciences, political science, distributed systems, robotics, and more recently, in aligning AI systems with human preferences. At its core, such decision-making involves taking into account multiple agents – individuals, autonomous systems, or organizations – each with their own objectives, preferences, and constraints, to make coherent and coordinated decisions within complex, dynamic environments. The complexity of decision-making in MAS arises from the interplay of several factors, including uncertainty, inherent biases, conflicting objectives, and the limitations of the agents' computational and observational capabilities.

Von Neumann & Morgenstern (1947) reformulated and popularized *expected utility theory* Ramsey (1926), laying the foundation for *rational* decision-making, where agents select actions to maximize their expected utility. Since then, Game Theory (GT) has become central to MAS, with models, such as Nash equilibrium Nash Jr (1950), cooperative game theory Shapley (1953), evolutionary game theory, and Bayesian games Harsanyi (1967), offering tools to analyze scenarios where agents' choices impact one another. Although rational decisions are grounded in systematic analysis and objective reasoning, human choices are often influenced by cognitive biases, emotions, social factors, and various unobserved factors that lead to seemingly irrational outcomes. Sometimes, irrational or naive choices can even result in better outcomes than rational ones, a phenomenon known as *paradox of rationality* Howard (1971); Colman (2003); Basu (1994). Behavioral economics seeks to model such deviations from rationality, with models such as loss aversion Kahneman & Tversky (1979), anchoring Tversky & Kahneman (1974), framing of choices Kahneman & Tversky (1984), social preferences Fehr & Schmidt (1999), and emotions Loewenstein (2003). Kahneman (2011) also advanced and popularized *dual-process theory* Wason & Evans (1974); Sloman (1996), which posits two cognitive systems: a fast automatic *System 1* and a slow deliberate *System 2*. While these

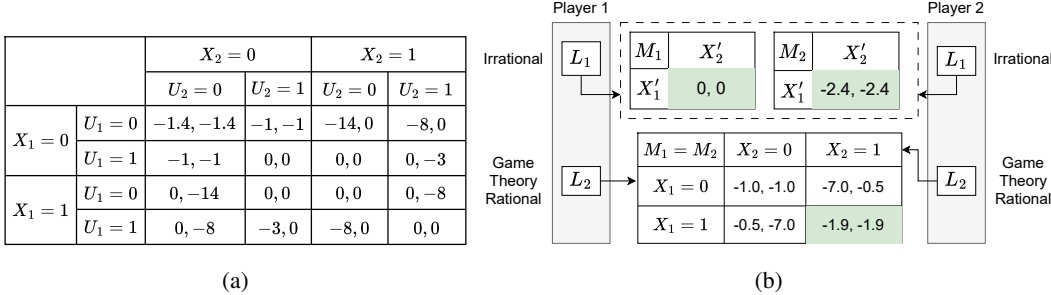

|  |  | $X_2 = 0$ |  | $X_2 = 1$ |  |
|---|---|---|---|---|---|
|  |  | $U_2 = 0$ | $U_2 = 1$ | $U_2 = 0$ | $U_2 = 1$ |
| $X_1 = 0$ | $U_1 = 0$ | $-1.4, -1.4$ | $-1, -1$ | $-14, 0$ | $-8, 0$ |
|  | $U_1 = 1$ | $-1, -1$ | $0, 0$ | $0, 0$ | $0, -3$ |
| $X_1 = 1$ | $U_1 = 0$ | $0, -14$ | $0, 0$ | $0, 0$ | $0, -8$ |
|  | $U_1 = 1$ | $0, -8$ | $-3, 0$ | $-8, 0$ | $0, 0$ |

(a)          (b)

Figure 1: (a) $Y_1, Y_2$ as a function of $U_1, U_2, X_1, X_2$ (b) Should the agents be rational or not?

approaches help explain aspects of irrational human decision-making, the broader question of *when* and *how* such unobserved biases can be strategically leveraged in MAS remains largely unexplored.

In this work, we make a significant step towards addressing this gap by proposing a framework rooted in causal modeling Pearl (2009); Bareinboim et al. (2022). Human decisions are often guided by causal structures Tversky & Kahneman (2015); Sloman & Hagmayer (2006); Nichols & Danks (2007), and actions can be viewed as interventions Hagmayer & Sloman (2009). Building on these insights, we model the environment and the agent's decision-making process as an interplay between exogenous and endogenous factors, represented as a structural causal model (SCM). SCMs have been used successfully in the context of decision-making, both for single-step bandit problems Bareinboim et al. (2015); Zhang & Bareinboim (2017) and for multistep RL settings Lee & Bareinboim (2020); Ruan et al. (2023), as surveyed in Bareinboim et al. (2024). The advantage of such modeling is not only computational but more fundamental. Consider the example of *Greedy Casino* in Bareinboim et al. (2015), where a randomized control trial (RCT) suggests that the expected payoff is higher than the realized payoff of players following their natural instincts (irrational behavior). One may naturally surmise that, given the superiority of the automated version based on RCTs, humans and their irrationality could be removed from the loop. However, players could enact a counterfactual randomization procedure that exploits their natural biases, which surprisingly led to payoffs exceeding those based on the RCT.

Building on these insights, we model MAS through a causal lens and show that existing game models may not capture similar fundamental features of the decision-making process. This framework models the interactions of agents within a system through the different layers of PCH Bareinboim et al. (2022). As a consequence, an agent will have the capability to act rationally (following Nash's prescription), instinctively, or as some mixture of both. We introduce the notion of *counterfactual rationality* to formally determine when it is advantageous for agents to act irrationally and when it is better to avoid doing so. The next example illustrates why this task is nontrivial.

**Example 1.1** (Causal Prisoner's Dilemma (CPD)). Two thieves are suspected of a crime, but due to insufficient evidence, they cannot be convicted outright. Now, they have a choice to make – either remain silent (cooperate, $C$) or betray the other (defect, $D$). We denote the choices by variables $X_1$ and $X_2$, and cooperation and defection by the values 0 and 1, respectively. The thieves' decisions are influenced by external circumstances, represented by variables $U_1$ and $U_2$, which capture factors such as the temperament of police officers, the competence of legal defense, new evidence or witnesses emerging, and even the disposition of the judge and the jury. Although these factors cannot be explicitly measured by the prisoners, they may subconsciously shape their decisions.

Each prisoner has a natural ability to assess their circumstances, denoted by $R_1$ and $R_2$. If prisoner $i$ has an accurate reading of their situation ($R_i = 1$), they choose to cooperate ($X_i = 0$) if the circumstances are favorable ($U_i = 1$), and defect when they are adversarial ($U_i = 0$); conversely, if they have a poor reading of their situation ($R_i = 0$), they defect when circumstances are good, and cooperate when circumstances are bad. For prisoner $i$, their instinctive or natural choice is modeled as: $X_i \leftarrow f_X(R_i, U_i) = R_i \oplus U_i$, where $\oplus$ is the exclusive-or operator. We note that the variables $U_1, U_2, R_1, R_2$ and the function $f_X$ are determined by nature and are unknown to the prisoners.

Now, we analyze two scenarios, $M_1$ and $M_2$. In $M_1$, the prisoners have a good reading of their situation ($R_1 = R_2 = 1$), while in $M_2$, they misjudge their circumstances ($R_1 = R_2 = 0$). In both

cases, $P(U_i = 0) = 0.6$ for $i \in \{1, 2\}$. The outcome $\mathbf{Y} = (Y_1, Y_2)$ of their decisions is a function of $U_1, U_2, X_1$ and $X_2$ as shown in Fig. 1a. For example, when the situation is favorable for both the prisoners ($U_1 = 1, U_2 = 1$) and they cooperate ($X_1 = 0, X_2 = 0$), their payoff is $(0, 0)$. However, if circumstances are favorable for Prisoner 1 and not for Prisoner 2 ($U_1 = 1, U_2 = 0$), and Prisoner 1 defects while Prisoner 2 cooperates ($X_1 = 1, X_2 = 0$), their payoff is $(0, -8)$.

If both prisoners ignore their intuition and search for the optimal strategy, the situation corresponds to the classical Prisoner's Dilemma, where the payoff for the actions $X_1 = x_1, X_2 = x_2$ is given by:

$$\sum_{u_1, u_2, \mathbf{y}} \mathbf{Y} \cdot P(u_1, u_2) P(\mathbf{Y} \mid x_1, x_2, u_1, u_2) \qquad (1)$$

Notably, both scenarios $M_1$ and $M_2$ lead to the same Prisoner's Dilemma (PD) game, as shown in the $2 \times 2$ payoff table at the bottom of Fig. 1b. However, if both prisoners rely on their natural instincts, their expected payoff is $(0, 0)$ in $M_1$ and approximately $(-2.4, -2.4)$ in $M_2$. This is illustrated in Fig. 1b, where $X_1'$ and $X_2'$ denote the players acting based on their natural intuition (shown in the top row). The situation presents a new dilemma – it is better to follow natural instincts and be irrational in $M_1$, whereas it is better to be rational and ignore intuition in $M_2$.

This example raises a fundamental question: when is it better to follow natural intuition and when is it better to override it and follow Nash's prescription? In this paper, we explore the tension between rational and instinctive behavior through a causal lens and derive from first principles how agents should deliberate and make decisions, thus addressing the so-called 'paradox of rationality' (see Appendix A). Specifically, we outline our technical contributions as follows:

1. We formalize a class of games that combine rational and irrational behavior (Def.2.10) and show that it strictly generalizes traditional Normal Form Games (Thm.2.11).

2. We introduce a new family of *counterfactual strategies*, prove the existence of equilibrium (Thm.3.5), and show that these strategies can outperform other strategies (Thm.3.6).

3. We develop an algorithm `CTF-Nash-Learning` (Alg. 2) that learns the payoff matrix in the counterfactual action space and identifies equilibria, even when the actions of the other agents are not fully observed.

**Preliminaries.** In this section, we introduce the notations and definitions used throughout the paper. We use capital letters to denote random variables ($X$) and small letters to denote their values ($x$). $\mathcal{D}_X$ denotes the domain of $X$. $|\mathbf{S}|$ denotes the cardinality of the set $\mathbf{S}$. The basic framework of our model resides on Structural Causal Models Pearl (2009). An SCM $M$ is a tuple $\langle \mathbf{U}, \mathbf{V}, \mathcal{F}, P(\mathbf{U}) \rangle$, where $\mathbf{V}$ and $\mathbf{U}$ are sets of endogenous and exogenous variables, respectively. $\mathcal{F}$ is a set of functions $f_V$ determining the value of $V \in \mathbf{V}$, that is, $V \leftarrow f_V(\mathbf{Pa}(V), \mathbf{U}_V)$, where $\mathbf{Pa}_V \subseteq \mathbf{V}$ and $\mathbf{U}_V \subseteq \mathbf{U}$. Naturally, $M$ induces a distribution over the endogenous variables, $P(\mathbf{V})$, called *observational or $L_1$ distribution*. An intervention on a subset $\mathbf{X} \subseteq \mathbf{V}$, denoted by $do(\mathbf{x})$ is an operation where values of $\mathbf{X}$ are set to $\mathbf{x}$, replacing the functions $\{f_X : X \in \mathbf{X}\}$. For an SCM $M$, $M_\mathbf{x}$ denotes the model induced by the operation $do(\mathbf{x})$ and $P_\mathbf{x}(\mathbf{Y})$ or $P(\mathbf{Y}_\mathbf{x})$ denotes the probability of $\mathbf{Y}$ in $M_\mathbf{x}$. Such distributions are called *interventional or $L_2$ distributions*. For further details and discussions on counterfactual distributions, refer to Appendix A.1 and Bareinboim et al. (2022, Sec.1.2). Additional background and examples on decision-making in single-agent causal systems can be found in Bareinboim et al. (2024) and Appendix A.5, along with comparisons to related work Hammond et al. (2023); Gonzalez-Soto et al. (2019) in AppendixA.

## 2 CAUSAL NORMAL FORM GAMES

In this section, we model the interaction of multiple agents in a system through the language of SCMs and PCH layers. Here, we generalize the concepts introduced in Bareinboim et al. (2024) to multi-agent settings. We first define a set of action nodes and reward signals for the agents in the system along with the SCM.

**Definition 2.1** (Causal Multi-Agent System). A Causal Multi-Agent System (CMAS) is a tuple $\langle M, N, \mathbf{X}, \mathbf{Y} \rangle$, where (i) $M : \langle \mathbf{U}, \mathbf{V}, \mathcal{F}, \mathbb{P} \rangle$ is an SCM, (ii) $N$ is the set of $n$ agents, (iii) $\mathbf{X} = (\mathbf{X}_1, \ldots, \mathbf{X}_n)$ is a tuple of action nodes with disjoint $\mathbf{X}_i, \mathbf{X}_j \subset \mathbf{V}$ for $i, j \in [n]$, $i \neq j$, and (iv) $\mathbf{Y} = (\mathbf{Y}_1, \ldots, \mathbf{Y}_n)$ is the ordered set of reward signals, with $\mathbf{Y}_i \subseteq \mathbf{V} \setminus \mathbf{X}$ for all $i \in [n]$. $\qquad \square$

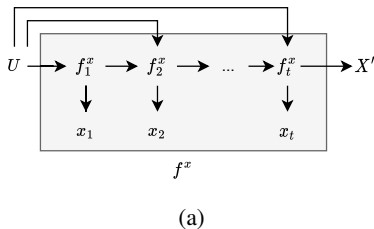

| | P2 | $X_2 = X_2'$ | $X_2 = 0$ | $X_2 = 1$ |
|---|---|---|---|---|
| P1 | | | | |
| $X_1 = X_1'$ | | $-2, 2$ | $-2, -2$ | $-2, -2$ |
| $X_1 = 0$ | | $0, 0$ | $-1.5, -1.5$ | $-1.5, -1.5$ |
| $X_1 = 1$ | | $0, 0$ | $-1.5, -1.5$ | $-1.5, -1.5$ |
| $X_1 = 1 - X_1'$ | | $2, -2$ | $-1, -1$ | $-1, -1$ |

(a)  (b)

Figure 2: (a) Illustration of decision flow $f_X$ (b) It is not always optimal to jump to $L_3$ policy

A CMAS is essentially an SCM with a set of action nodes $\mathbf{X}$, each controlled by one of the $n$ agents. In addition, the system includes reward variables, $\mathbf{Y}$, representing the feedback each agent receives based on their actions and the underlying causal mechanism.

**Example 2.2.** Consider the CPD presented in Ex. 1.1. The SCM $\mathcal{M}$ corresponding to scenario $M_2$ is defined as: (i) $\mathbf{U} = \{U_1, U_2, R_1, R_2\}$, $\mathbf{V} = \{X_1, X_2, Y_1, Y_2\}$, (ii) $X_i = R_i \oplus U_i$ for $i \in \{1, 2\}$. $Y_1, Y_2$ as a function of $U_1, U_2, X_1, X_2$ are shown in Fig. 1a, and (iii) $P(U_i = 1) = 0.4, P(R_i = 0) = 1$ for $i \in \{1, 2\}$. The CMAS can now be defined as $\langle M = \mathcal{M}, N = \{1, 2\}, \mathbf{X} = (\{X_1\}, \{X_2\}), \mathbf{Y} = (\{Y_1\}, \{Y_2\})$. $\qquad\square$

Now, we define different forms of actions that an agent may take in such a system. First, we define the different action and policy spaces and then explore how the action spaces are related.

**Definition 2.3** ($L_1$ action). Given a CMAS $\langle M, N, \mathbf{X}, \mathbf{Y} \rangle$, an $L_1$ action of an agent $i$ is the one in which the value of their action variables $\mathbf{X}_i$ is determined by the natural mechanism $f_{\mathbf{X}_i} \in \mathcal{F}$. $\qquad\square$

We will also call such actions *natural actions* and denote them by $a_0$. Note that, while performing $a_0$, an agent *does not know anything about the underlying SCM* nor do they deliberately change any mechanism of action variable in the system. The $L_1$ action space is thus $\mathcal{A}^1 = \{a_0\}$ and the $L_1$ policy space is also a singleton set $\Pi^1 = \{a_0\}$.

**Example 2.4.** Consider the CMAS presented in Ex. 2.2. The natural action is when the values of $X_1$ and $X_2$ are determined by their natural function, $X_1 = R_1 \oplus U_1, X_2 = R_2 \oplus U_2$ The expected payoff when both the agents are following their natural intuition is then given by

$$\sum_{u_1, u_2, x_1, x_2, \mathbf{y}} \mathbf{y} \cdot P(u_1, u_2) P(x_1 \mid u_1) P(x_2 \mid u_2) P(\mathbf{y} \mid u_1, u_2, x_1, x_2) \approx (-2.4, -2.4) \quad (2)$$

In traditional game-theoretic sense, an agent can intervene on the system via atomic interventions (setting action variables to fixed values based on context) Pearl (2009), or soft interventions (sampling actions from a distribution) Correa & Bareinboim (2020). Next, we define $L_2$ actions and the associated policy space.

**Definition 2.5** ($L_2$-action). Given a CMAS $\langle M, N, \mathbf{X}, \mathbf{Y} \rangle$, $L_2$ action of an agent $i$ is a hard intervention $do(\mathbf{x})$, where $\mathbf{x} \in \mathcal{D}_{\mathbf{X}_i}$. $\qquad\square$

Hence, if an agent $i$ performs $do(\mathbf{x}_i)$ in the SCM $M$, then the natural mechanism $f_{\mathbf{X}_i}$ is replaced by $\mathbf{X}_i \leftarrow \mathbf{x}_i$. The set of such $L_2$ actions is denoted by $\mathcal{A}^2$, and an $L_2$ policy is a distribution over $\mathcal{A}^2$.

**Example 2.6.** Consider the CMAS introduced in Ex. 2.2. $L_2$ action is when an agent performs an intervention, that is, setting their action variable to a particular value. If Player 1 is playing 0 and Player 2 is playing 1, then the assignment of the variables are given by $X_1 \leftarrow 0, X_2 \leftarrow 1$ and $U_1, U_2, R_1, R_2$ are sampled from $P(\mathbf{U})$ as in Ex. 2.2. Similarly, $Y_1, Y_2$ are determined by Fig. 1a. For instance, the expected payoff of the strategy $(do(X_1 = 0), do(X_2 = 1))$ will then be given by

$$\sum_{u_1, u_2, \mathbf{y}} \mathbf{y} \cdot P(u_1, u_2) P(\mathbf{y} \mid u_1, u_2, X_1 = 0, X_2 = 1) \approx (-7.0, -0.5) \quad (3)$$

It is also possible for one agent to perform an $L_2$ action and the other to perform an $L_1$ action. For instance, the payoff the strategy $(do(X_1 = 1), a_0)$ is given by

$$\sum_{u_1, u_2, x_2, \mathbf{y}} \mathbf{y} \cdot P(u_1, u_2) P(x_2 \mid u_2) P(\mathbf{y} \mid u_1, u_2, X_1 = 1, x_2) \approx (0, -8.9) \quad (4)$$

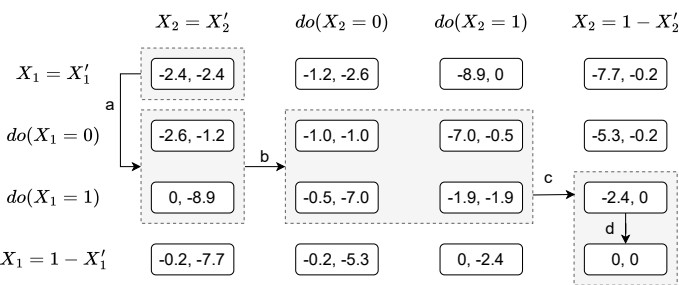

Figure 3: Change of Equilibrium with change of policies in Causal Prisoner's Dilemma.

In many cases, an agent can interact with the environment through PCH's Layer 3 Bareinboim et al. (2015; 2022); Raghavan & Bareinboim (2025a), enabling counterfactual reasoning in their decision-making, and entering the realm of $L_3$ distribution. For example, in scenario $M_2$ of Ex. 1.1, following natural instinct led to a suboptimal outcome. However, if both agents had done the exact opposite of their instinctive choices, they could have achieved a payoff of $(0, 0)$. Now, we formally define $L_3$ actions.

**Definition 2.7** ($L_3$-action space $\mathcal{A}^3$). Given a CMAS $\langle N, M, \mathbf{X}, \mathbf{Y} \rangle$, an $L_3$ action for agent $i$ is defined as a mapping $h : \mathcal{D}(\mathbf{X}_i) \to \mathcal{D}(\mathbf{X}_i)$ from intuition to action. $\square$

When an agent takes an $L_3$ action, they first note their natural instinct $\mathbf{X}'_i \leftarrow f_{\mathbf{X}_i}(\mathbf{U}_i)$ and then executes $\mathbf{X}_i \leftarrow h_i(\mathbf{X}'_i)$, where $\mathbf{U}_i$ is the set of unobserved parents of $\mathbf{X}_i$. If $h(x) = x$, it corresponds to the $L_1$ action, and if $h(x)$ is constant for all $x$, it is the $L_2$ action. Bearing this in mind, we will often denote $a_0$ as $\mathbf{X} = \mathbf{X}'$, where $\mathbf{X}$ is the action variable and $\mathbf{X}'$ is the intuition.

Bareinboim et al. (2015) introduces a novel form of randomization to interact through the Layer 3 of PCH – interrupt any reasoning agent just before they execute their choice, treat this choice as their intention, and then act. This procedure involves subtle issues, and we refer readers to Sec. 7 in Bareinboim et al. (2024) for a more detailed discussion. The agent may consider various options during the deliberation process, but only the final choice matters. For example, an agent may initially choose $X' = x_1$, then reconsider and change it to $X' = x_2$ and may continue doing so, until at time step $t$, it chooses $X' = x_t$ and decides to execute it. This final decision defines the agent's instinct irrespective of the path taken to reach it (see Fig. 2a). The same reference also proposed Ctf-RCT, where an intended action is perceived first, but instead of executing it directly, the final action is chosen uniformly at random from the entire action space. Now, we look at how to compute the payoff under $L_3$ action.

**Example 2.8.** Consider the CMAS in Ex. 2.2. An $L_3$ action would allow the agent to choose an action based on their natural intuition. Let $g_1$ and $g_2$ be two functions from $\{0, 1\}$ to $\{0, 1\}$. If the players are playing $g_1$ and $g_2$, respectively, then the variables are given by $X'_i \leftarrow R_i \oplus U_i$, $X_i \leftarrow g_i(X'_i)$ for $i \in \{1, 2\}$. The variables $U_1, U_2, R_1, R_2$ are sampled from $P(\mathbf{U})$, and $Y_1, Y_2$ are determined by Fig. 1a. For example, if $g_1(x) = 1 - x$ and $g_2(x) = 1 - x$, then the expected payoffs are given by

$$\sum_{u_1, u_2, x_1, x_2, \mathbf{y}} \mathbf{y} \cdot P(u_1, u_2)P(x_1, x_2 \mid u_1, u_2)P(\mathbf{y} \mid u_1, u_2, g_1(x_1), g_2(x_2)) = (0, 0) \quad (5)$$

The payoffs for the various combinations of actions in Ex. 2.2 are shown in Fig. 3. Once the action spaces are defined, the policy space can be defined as a distribution over the action space. Let $\Delta(A)$ denote the set of distributions over the set of actions $A$. Then $L_2$ policy space $\Pi^2 = \Delta(\mathcal{A}^2)$ and $L_3$ policy space $\Pi^3 = \Delta(\mathcal{A}^3)$. Next, we define the notion of reward.

**Definition 2.9** (Reward Function). A reward function $\mathcal{R}_i : \mathcal{D}(\mathbf{Y}_i) \to \mathbb{R}$ of an agent $i$ is a function from outcome $\mathbf{Y}_i$ to real numbers. $\square$

In Ex. 1.1, we assume that the reward function is identity, that is, $\mathcal{R}_i(Y_i) = Y_i$ for $i \in \{1, 2\}$. Now that we have all the tools, we are ready to define Normal Form Games in proper causal language.

**Definition 2.10** (Causal Normal Form Game). A tuple $\Gamma = \langle \mathbb{M}, \mathcal{A}, \mathcal{R} \rangle$ is a Causal Normal Form Game (CNFG), where (i) $\mathbb{M}$ is a CMAS $\langle M, N, \mathbf{X}, \mathbf{Y} \rangle$, (ii) $\mathcal{A} = (\mathcal{A}_1, \dots, \mathcal{A}_n)$ is the set of policies for the $n$ agents, $\mathcal{A}_i \subseteq \{\mathcal{A}^1, \mathcal{A}^2, \mathcal{A}^3\}$, and (iii) $\mathcal{R} = (\mathcal{R}_1, \dots, \mathcal{R}_n)$ is the set of reward functions. $\square$

A CNFG is thus a CMAS, along with the policy space of the $n$ agents and their reward functions. Now we will formally state the result generalizing our observation from CPD (Ex. 1.1).

**Theorem 2.11.** *Given a game in normal form, there exist two CNFGs $\mathcal{C}_1$ and $\mathcal{C}_2$ with equilibrium payoffs $\mu_1$ and $\mu_2$ under the action space $\mathcal{A}^1 \cup \mathcal{A}^2$, and a Nash Equilibrium (NE) payoff $\mu_{NE}$, such that $\mu_2 < \mu_{NE} < \mu_1$ where $<$ denotes Pareto domination.* $\square$

The theorem implies some important observations. CNFGs strictly generalize Normal Form Games (NFGs), capturing aspects such as instinctive behaviors and counterfactual policies that NFGs cannot naturally express. Although one might argue that CNFGs can be flattened into an equivalent NFG (Fig. 3), similar to Extensive Form or Bayesian Games, we claim causal modeling is not only advantageous but necessary: (i) Constructing the full payoff matrix requires an SCM, since actions are not arbitrary and defined only within that causal structure; (ii) NFGs do not clarify how actions are executed or whether agents are even capable of executing them; SCMs provide a concrete notion of agency; (iii) our solution concept presented in Sec. 3 relies on the hierarchical structure of the action spaces; (iv) finally, NFGs cannot capture the structure between intuitions and executed actions. In many cases, agents can only observe executed actions, and for computing equilibria, exploiting the structure becomes a necessity (Alg. 2). More details are provided in Appendix E.

## 3 CAUSAL NASH EQUILIBRIUM

In this section, we introduce counterfactual rationality and establish the Causal Nash Equilibrium (CNE) for a CNFG. Allowing agents to transition between layers of the PCH leads to a two-step decision process. First, the agent determines which layers to operate in – instinct-based ($L_1$), classical rationality ($L_2$), or counterfactual reasoning ($L_3$). Second, the agent must decide which action to take within the chosen layer. We refer to this two-step process as a *causal strategy*. An agent is counterfactually rational if it seeks to maximize its expected payoff using causal strategies, given that other agents are also counterfactually rational. Next, we analyze how equilibrium outcomes change when agents move to higher layers of the PCH.

**Example 3.1** (Equilibria in CPD). Consider Ex. 1.1 ($M_2$) where we analyze how the payoffs and equilibria evolve as agents move across the layers of the PCH, from instinct-based $L_1$ policies to counterfactual-based, $L_3$ policies. Fig. 3 shows the payoff of the prisoners in this larger action space. If both prisoners follow their natural choices, playing $L_1$, their payoffs are $(-2.4, -2.4)$.

Now, suppose prisoner 1 starts thinking rationally, ignoring their natural instincts, which results in transition (a) in Fig. 3. Prisoner 1 eventually defects, meaning they play the action $do(X_1 = 1)$, while prisoner 2 still follows their instinct, $X_2' = X_2$. As a result, the payoffs become $(0, -8.9)$, where prisoner 1 benefits while prisoner 2 suffers. Eventually, prisoner 2 also learns to think rationally, leading to transition (b). In this case, both prisoners enter the realm of Standard Game Theory (SGT), each choosing to defect, playing the actions $(do(X_1 = 1), do(X_2 = 1))$. This results in NE with payoffs of $(-1.9, -1.9)$. A few observations are worth making at this point. First, the scope of SGT is highlighted in the four central cells of Fig. 3. Second, as noted earlier, the equilibrium in SGT is worse than when both agents act irrationally ($L_1$). The SGT analysis stops at this point, but our new framework suggests that strategic thinking may continue.

Over time, prisoner 2 introspects and contemplates counterfactual decisions, as highlighted in transition (c). They realize that their natural instincts provide insights that can be leveraged, and they should choose to act opposite to their natural choices, $X_1 = 1 - X_1'$. This yields payoffs of $(-2.4, 0)$, improving their baseline and hurting prisoner 1. Eventually, prisoner 1 also reaches $L_3$, leading to transition (d). Both players, now operating under Causal Game Theory (CGT), settle on actions against their natural instincts, $X_1 = 1 - X_1', X_2 = 1 - X_2'$, achieving payoffs of $(0, 0)$. This is the final state, where no unilateral deviation can increase payoffs. $\square$

The game in this example reflects an increasingly refined form of human rationality, tracing its evolution from primitive instincts based on raw intuition ($L_1$) to a notion of rationality based on game theory, where the intuition is ignored ($L_2$), and going to advanced strategic thinking leveraging both rational and irrational aspects of human cognition ($L_3$). A natural question that arises from this discussion is if it is always better to consider the full payoff table, since it provides the largest action space. To answer this, consider the example shown in Fig. 2b. The full game specification is given in Appendix D. If Player 1's action space is limited to $L_1$ and $L_2$, then the equilibrium payoff is

| P2 \ P1 | $\mathcal{A}^1$ | $\mathcal{A}^2$ | $\mathcal{A}^1 \cup \mathcal{A}^2$ |
|---|---|---|---|
| $\mathcal{A}^1$ | $-2, 2$ | $-2, -2$ | $-2, 2$ |
| $\mathcal{A}^2$ | $0, 0$ | $-1.5, -1.5$ | $0, 0$ |
| $\mathcal{A}^1 \cup \mathcal{A}^2$ | $0, 0$ | $-1.5, -1.5$ | $0, 0$ |
| $\mathcal{A}^3$ | $2, -2$ | $-1, -1$ | $-1, -1$ |

(a)

| P2 \ P1 | $\mathcal{A}^1$ | $\mathcal{A}^2$ | $\mathcal{A}^1 \cup \mathcal{A}^2$ | $\mathcal{A}^3$ |
|---|---|---|---|---|
| $\mathcal{A}^1$ | $-2.4, -2.4$ | $-8.9, 0$ | $-8.9, 0$ | $-8.9, 0$ |
| $\mathcal{A}^2$ | $0, -8.9$ | $-1.9, -1.9$ | $-1.9, -1.9$ | $-2.4, 0$ |
| $\mathcal{A}^1 \cup \mathcal{A}^2$ | $0, -8.9$ | $-1.9, -1.9$ | $-1.9, -1.9$ | $-2.4, 0$ |
| $\mathcal{A}^3$ | $0, -8.9$ | $0, -2.4$ | $0, -2.4$ | $0, 0$ |

(b)

Figure 4: Layer selection game for (a) example in Fig. 2b, and (b) Causal Prisoner's Dilemma.

$(0, 0)$ (marked in blue). However, if the action space $L_3$ is considered, the last row in the table is also considered (gray), and the equilibrium payoff decreases to $(-1, -1)$. Hence, regardless of what the other player does, Player 1's mere consideration of a larger action space harms them. Broadly, deciding which action space to follow is non-trivial. Next, we define a projection of a CNFG, where action spaces are restricted to specific layers of the PCH.

**Definition 3.2** (PCH Projection). Given a CNFG $\Gamma = \langle \mathbb{M}, \mathcal{A}, \mathcal{R} \rangle$, the PCH projection of $\Gamma$, denoted by $\Gamma(A_1, \ldots, A_n)$, is the subgame of $\Gamma$ where the action space of agent $i$ is constrained to a subset $\mathcal{A}_i \supseteq A_i \in \{\mathcal{A}_i^1, \mathcal{A}_i^2, \mathcal{A}_i^1 \cup \mathcal{A}_i^2, \mathcal{A}_i^3\}$. □

This projection captures how a game evolves when agents operate within a restricted subset of available strategies corresponding to different levels of reasoning within the PCH. The problem now, is to find a projection from where agents have no incentive to unilaterally deviate to a different layer of the PCH. To address this, we introduce a strategic layer selection game, a meta-game, where agents choose which layer of PCH to operate at.

**Definition 3.3** (Layer Selection Game). Given a CNFG $\Gamma = \langle \mathbb{M}, \mathcal{A}, \mathcal{R} \rangle$, its Layer Selection Game $L_\Gamma$ is the NFG with (i) the same set of agents $N$, (ii) action space $A = A_1 \times \ldots, A_n$, where $\mathcal{A}_i \supseteq A_i \in \{\mathcal{A}_i^1, \mathcal{A}_i^2, \mathcal{A}_i^1 \cup \mathcal{A}_i^2, \mathcal{A}_i^3\}$ and (iii) utility $u(A) = \text{NE}(\Gamma(A_1, \ldots, A_n))$ where $\text{NE}(\Gamma(A_1, \ldots A_n))$ is a Nash Equilibrium payoff of the CNFG $\Gamma$ when actions spaces are restricted to $A_1, \ldots, A_n$. □

This metagame represents a higher-level decision process, where each cell in the payoff matrix corresponds to a PCH projection of $\Gamma$, and its equilibrium will determine the layer of reasoning in which the agents should operate. We will assume that such counterfactual rationality is common knowledge, that both players are aware that the other player can forget a part of their actions space and choose the PCH layers in which they operate. Let $s_i^*$ be the NE strategy for player $i$ in the layer selection game. Let $\text{supp}(s_i^*)$ denote the support of $s_i^*$ – the set of action spaces with non-zero probability in $s_i^*$. In particular, if $\mathcal{A}_i^j \notin \text{supp}(s_i^*)$, then the agent can ignore, or "forget" about this action space, and instead play a PCH projection of $\Gamma$ that excludes $\mathcal{A}_i^j$. For instance, in Fig. 2b, if Player 1 is able to forget that it can play $L_3$, the payoff for the agent is $(0, 0)$, which is higher than the payoff that with playing $L_3$, $(-1, -1)$.

In practice, agents can limit their reasoning layers by restricting their capabilities: (i) at $L_1$, agents act instinctively without requiring sampling mechanisms, (ii) at $L_2$, agents may need access to randomization (e.g., coin flips) for mixed strategies, and (iii) At $L_3$, agents must introspect, observe their intuition, and then decide how to act based on it, through more sophisticated procedures, such as ctf-randomization. Refusing to observe intuition renders $L_3$ inaccessible. One key observation is that forgetting part of the action space may not always be a good idea. For example, consider the simple prisoner's dilemma. If the agents choose to forget defect $D$ and just play with the action space cooperate $\{C\}$, they will get a payoff $(-1, -1)$. However, one agent may start using the action space $\{C, D\}$ and then choose to defect, obtaining a payoff of $-0.5$ while the other agent gets $-7.0$. Thus it is not in the agent's interest to forget about defecting (see Appendix D).

**Definition 3.4** (Causal Nash Equilibrium, or CNE). Let $\Gamma$ be a CNFG and $L_\Gamma$ be its corresponding layer selection game with NE strategy $s^*$. A strategy profile $\pi^*$ is called CNE if $\pi^*$ is the Nash Equilibrium of $\Gamma(A^*)$, where $A^* = A_1 \times \ldots \times A_n$, and $A_i = \cup_{\mathcal{A} \in \text{supp}(s_i^*)} \mathcal{A}$. □

**Theorem 3.5** (Existence of CNE). *For any CNFG, CNE always exists.* □

If playing $L_2$ is a pure strategy NE of the layer selection game $L_\Gamma$, then the CNE of $\Gamma$ in CGT and the NE of the normal form game induced by $\Gamma$ coincide. Note that it is possible for a CNFG to have

---

**Algorithm 1** Find-CNE

---

1: **Input:** PCH projections of CNFG $\Gamma = \langle \mathbb{IM}, \mathcal{A}, \mathcal{R} \rangle$ **Output:** CNE strategies $\pi^*$
2: Construct the Layer Selection Game, $L_\Gamma$: For all $A = A_1 \times \ldots \times A_n$, such that $\mathcal{A}_i \supseteq A_i \in \{\mathcal{A}^1, \mathcal{A}^2, \mathcal{A}^1 \cup \mathcal{A}^2, \mathcal{A}^3\}$, $u(A) \in \text{NE}(\Gamma(A_1, \ldots, A_n))$
3: Let $s^*$ be the NE strategy of $L_\Gamma$ and $A^* = A_1^* \times \ldots A_n^*$, where $A_i^* = \bigcup_{\mathcal{A} \in \text{supp}(s_i^*)} \mathcal{A}$
4: **Return:** NE strategies of $\Gamma(A^*)$

---

**Algorithm 2** Ctf-Nash-Learning

---

1: **Input:** Dataset from Ctf-RCT: $(x_1', x_1, x_2, \mathbf{y})$
2: **Output:** Causal Nash Equilibrium strategy $f^*$
3: For each $(x_1', x_1, x_2)$, estimate the mean and weights of the distributions' mixture from the samples $(y_1, y_2)$. Let the distribution means be $R_1(x_1', x_1, x_2), \ldots, R_k(x_1', x_1, x_2)$ with corresponding weights $p_1(x_1', x_1, x_2), \ldots, p_k(x_1', x_1, x_2)$ (in descending order)
4: If $k$ distributions cannot be identified, assume they are from a single distribution set $R_i(x_1', x_1, x_2)$ as the mean of the distribution and $p_i(x_1', x_1, x_2) = p_i(x_1', \bar{x}_1, \bar{x}_2)$ where $x_1, x_2 \neq \bar{x}_1, \bar{x}_2$. In case this assignment fails, set $p_i = 1/k$ for all $k$.
5: Define the action space for each player: $\mathcal{F}_1 = \{f : X_1' \to X_1\}$, $\mathcal{F}_2 = \{g : [k] \to X_2\}$
6: Construct a payoff matrix where each cell corresponds to a pair of functions $(f, g) \in \mathcal{F}_1 \times \mathcal{F}_2$. For each pair $(f, g)$, compute the payoff $\sum_{X_1', i} P(X_1') p_i(x_1', f(x_1'), g(i)) R_i(x_1', f(x_1'), g(i))$
7: $(f^*, g^*) \leftarrow$ Find-CNE on constructed payoff matrix without the action spaces $\mathcal{A}_2^1, \mathcal{A}_2^1 \cup \mathcal{A}_2^2$
8: **Return:** Strategy $f^*$.

---

multiple layer selection games and CNEs. Next, we look at how causal strategies compare with other strategies. $\text{NE}(\Gamma(A); L_\Gamma)$ is the NE payoff with action space $A$ as chosen in $L_\Gamma$.

**Theorem 3.6** (Dominance of causal strategies). *Let $\Gamma$ be a CNFG with CNE payoff $\mu^*$ and $L_\Gamma$ be its layer selection game with NE strategy $s^*$. If $s^*$ is a pure strategy NE and $A_i^* = supp(s_i^*)$, $\mu^* \geq NE(\Gamma(A_i, A_{-i}^*); L_\Gamma)$ for all $A_i \in \{A_i^1, A_i^2, A_i^1 \cup A_i^2, A_i^3\}$ and $i \in [n]$.* $\square$

In other words, Thm. 3.6 guarantees that if the layer selection game $L_\Gamma$ admits a pure strategy NE, no agent benefits by unilaterally switching to a different PCH reasoning layer. Consider Fig. 4a, which shows the layer selection game for the game in Fig. 2b: if Player 1 follows $L_2$ policies and Player 2 follows $L_1$ and $L_2$ policies, neither has an incentive to switch to a different layer of PCH. This leads to an interesting insight: CNE payoff of $\Gamma$ is thus $(0, 0)$, while the NE payoff of $\Gamma$ with $L_3$ actions is $(-1.5, -1.5)$ and that with interventions is $(0, 0)$. In contrast, Fig. 4b, corresponding to the CPD in Ex.3.1, has a pure strategy NE at $(\mathcal{A}^3, \mathcal{A}^3)$, indicating both players should adopt $L_3$ policies. This is consistent with Fig.3 resulting in a payoff $(0, 0)$ while NE payoff in $L_2$ is $(-1.9, -1.9)$.

## 4 LEARNING CAUSAL NASH EQUILIBRIUM

In this section, we introduce two algorithms for computing the CNE in CNFGs. First, we present Find-CNE (Alg. 1), which applies when the payoff matrix is common knowledge, as in SGT. Then, we propose Ctf-Nash-Learning, which learns the payoff matrix under partial observability.

We begin with the setting where the action spaces and corresponding payoffs of the CNFG $\Gamma$ are known to both agents (as in SGT). For example, if Player 1 has access to $L_3$ and Player 2 to $L_2$, both are aware of the payoffs for all combinations of actions within those spaces. We introduce Find-CNE (Alg. 1), which implements the ideas presented in Sec. 3. The algorithm first constructs the layer selection game $L_\Gamma$ corresponding to $\Gamma$ (step 2). and then computes its NE strategy (step 3). Any action space that occurs with nonzero probability in the NE strategy is used for CNE, or else discarded. Step 4 computes the NE of the projection of $\Gamma$ with the restricted action space.

However, such game dynamics may not be common knowledge. If the agents are learning the payoff matrix through exploration, they may be able to observe only the other agents' executed actions, but not their intuitions. To this end, we propose Ctf-Nash-Learning (Alg. 2), an algorithm that learns the payoff matrix in two-player CNFGs, where both agents have access to $L_3$ policy space. We assume that during exploration or learning phase, both players are playing Ctf-RCT Bareinboim

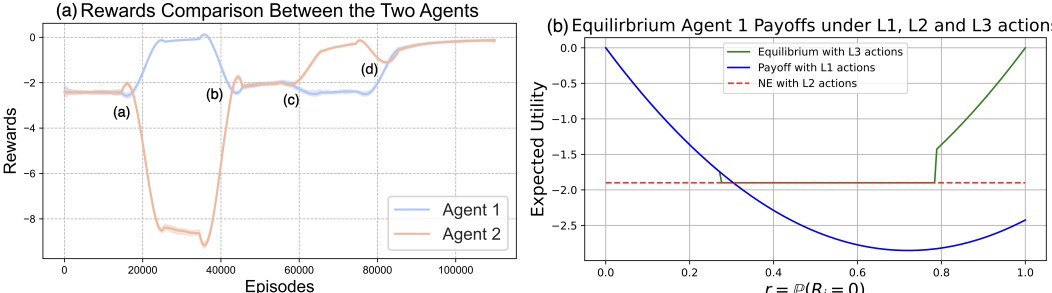

Figure 5: (a) Change in payoffs of the players in Causal Prisoner's Dilemma move up the layers of PCH. Transitions (a), (b), (c) and (d) corresponds to the ones indicated in Fig. 3 (b) Equilibrium Player 1 Payoffs with $L_1, L_2$ and $L_3$ action spaces under two conditions.

et al. (2024) and collect the dataset $(x'_1, x_1, x_2, \mathbf{y})$ (for player 1), where $x'_1$ is the intuition of player 1, $x_1$ and $x_2$ are the actions executed by players 1 and 2, respectively, and $\mathbf{y}$ is the reward tuple. The agents do not know the solution to the layer selection game or the optimal layer in which to play. For a fixed $(x'_1, x_1, x_2)$, the outcome $\mathbf{y}$ is sampled from the mixture $\sum_{x'_2} P(x'_2 \mid x'_1)P(\mathbf{y}_{x_1,x_2} \mid x'_2, x'_1)$. Step 3 recovers the means and weights of the mixture, which correspond (up to permutation) to $P(x'_2 \mid x'_1)$ and $E[\mathbf{Y}_{x_1,x_2} \mid x'_1, x'_2]$. In the CPD example, we identify $p_1(x'_1, x_1, x_2) \approx 0.6$ and $p_2(x'_1, x_1, x_2) \approx 0.4$ for all $(x'_1, x_1, x_2)$, matching $P(U_1 = 0)$ and $P(U_1 = 1)$. Examples of sample means include $R_1(0,0,0) = (-1.5, -1.5)$ and $R_2(0,0,0) = (-1,-1)$, corresponding to expectations conditioned on $X'_2 = 0$ and $X'_2 = 1$, respectively. These values can be consistently identified under certain technical assumptions (Appendix D). Step 4 addresses the degenerate cases where $\mathbf{Y}$ does not vary with intuition. Step 5 defines the agents' $L_3$ action spaces. In CPD, for agent $i$, it is $\{f(x) = x, f(x) = 0, f(x) = 1, f(x) = 1 - x\}$ corresponding to actions $\{X_i = X'_i, do(X_i = 0), do(X_i = 1), X_i = 1 - X'_i\}$. However, the other agents' intuitions deduced in this manner may be a permutation of the actual intuitions $X'_2$. Once we have a proxy for the $L_3$ actions, the payoff matrix can be computed using Step 6 and the CNE strategy using Find-CNE. The learned probabilities, mean, and payoff matrix for CPD are shown in the Appendix D.

**Theorem 4.1.** *Given a two player CNFG* $\Gamma = \langle \mathbb{M}, (\mathcal{A}^3_1, \mathcal{A}^3_2), \mathcal{R} \rangle$, *let $s^*$ be the NE strategy of the corresponding PCH-LSG* $L_\Gamma$ *and* $A_2 = \bigcup_{\mathcal{A} \in supp(s^*_2)} \mathcal{A}$. *If $A_2 \in \{\mathcal{A}^2_2, \mathcal{A}^3_2\}$, then* Ctf-Nash-Learning *correctly learns the CNE strategy for Player 1.* $\square$

**Experimental evaluation:** We empirically investigate how the behavior of the game changes when the players move across the layers of PCH. In order to simulate two agents learning, we enable them with Independent Q-Learning Tan (1993), a popular multi-agent RL algorithm. The dynamics as Player 1 moves up the layers of PCH, while Player 2 remains in the previous layer is shown in Fig. 5a This is an experimental realization of the discussions presented in Ex. 3.1 and Fig. 3. Every 20,000 timesteps, one of the agents moves up the layers of PCH, which triggers a change in payoff. Next, we also investigate how the equilibrium payoffs change with the value of $P(R_i = 0)$ for agent $i$ in Ex. 1.1. Earlier, we showed two extreme cases when $P(R_i = 0)$ is 0 and 1. we show the equilibrium payoffs for different values of $P(R_i = 0) = r$ for $i \in \{1, 2\}$. Note that, for the causal prisoner's dilemma, following $L_3$ policy space is better than following only $L_2$ action space.

## 5 CONCLUSIONS

In this work, we examine the tension between rational and irrational decision-making through a causal lens. We introduce an example where rationality is optimal in one setting and being instinctive in another, despite both yielding the same game-theoretic solution. To address this dilemma, we propose a causal framework that captures both rational and instinctive behaviors and strictly generalizes Normal Form Games (Thm.2.11). We define counterfactual strategies and analyze equilibrium properties under these strategies (Thm.3.6). Finally, we develop algorithms to compute such equilibria: Alg. 1 (known payoffs) and Alg. 2 (learning through interaction). We hope that this framework advances the design of more robust, rational decision-making systems.

# 6 ETHICS STATEMENT

The research was conducted in full compliance with the ICLR Code of Ethics, and the authors declare no conflicts of interest, sponsorship concerns, or ethical issues pertaining to the integrity of the results. This work does not involve human subjects, sensitive data, or experiments that pose potential harm to individuals, communities, or the environment. It does not present methodologies or insights with foreseeable malicious applications, nor does it introduce risks related to fairness, bias, discrimination, or privacy.

# 7 REPRODUCIBILITY STATEMENT

We have taken careful steps to ensure the reproducibility of our work. All theoretical results, including Theorems 2.11, 3.5, 3.6, and 4.1, are supported by complete proofs in Appendix B. Details of the experimental setup, along with the code used to generate the results and plots, are provided in Appendix D.3. All assumptions underlying our framework are explicitly stated in the main text and further elaborated in the appendix.

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
