# Supplementary Material

## A PRELIMINARIES AND BACKGROUND

### A.1 STRUCTURAL CAUSAL MODELS AND PCH

Structural Causal Models is a general class of data-generating models Pearl (2009); Bareinboim et al. (2024) that allows three types of distributions based on three levels of interaction with the system: observational, interventional, and counterfactual. First, we will give the formal definitions of these concepts and the hierarchical relation among them, known as Pearl Causal Hierarchy (PCH). Our presentation mostly follows Bareinboim et al. (2022).

**Definition A.1** (Structural Causal Models). A structural causal model $\mathcal{M}$ is a 4-tuple $\langle \mathbf{U}, \mathbf{V}, \mathcal{F}, P(\mathbf{U}) \rangle$, where

- $\mathbf{U}$ is a set of background variables, also called exogenous variables, that are determined by factors outside the model;

- $\mathbf{V}$ is a set $\{V_1, V_2, \ldots, V_n\}$ of variables, called endogenous, that are determined by other variables in the model — that is, variables in $\mathbf{U} \cup \mathbf{V}$.

- $\mathcal{F}$ is the set of functions $\{f_1, f_2, \ldots, f_n\}$ such that each $f_i$ is a mapping from (the respective domains of) $U_i \cup Pa_i$ to $V_i$, where $U_i \subset \mathbf{U}$, $Pa_i \subseteq \mathbf{V} \setminus V_i$, and the entire set $\mathcal{F}$ forms a mapping from $\mathbf{U}$ to $\mathbf{V}$, that is for each $i = 1, 2, \ldots, n$, we have $v_i \leftarrow f_i(pa_i, u_i)$;

- $P(\mathbf{U})$ is the distribution over $\mathbf{U}$.

One way to visualize the dependence among the variables in the SCM is through a causal diagram, formal construction of which is given below (Def. 13, Bareinboim et al. (2022) ).

**Definition A.2** (Causal Diagram (Semi-Markovian Models)). Given an SCM $\mathcal{M} = \langle \mathbf{U}, \mathbf{V}, \mathcal{F}, P(\mathbf{U}) \rangle$, a causal diagram $G$ of $\mathcal{M}$ is constructed as follows:

1. add a vertex for every endogenous variable in the set $\mathbf{V}$

2. add an edge $(V_i \rightarrow V_j)$, for every $V_i, V_j \in \mathbf{V}$ and $V_i$ occurs as an argument in $f_j \in \mathcal{F}$.

3. add a bidirected edge $(V_i \leftarrow \ldots \rightarrow V_j)$ for every $V_i, V_j \in \mathbf{V}$ if the corresponding $U_i, U_j \in \mathbf{U}$ are correlated or the corresponding functions $f_i, f_j$ share some $U \in \mathbf{U}$ as an argument.

Next, we define three types of distributions corresponding to distinct modes of interaction with an SCM: the $L_1$ (observational), $L_2$ (interventional), and $L_3$ (counterfactual) distributions (Defs. 2, 5, and 7 in Bareinboim et al. (2022)).

**Definition A.3** ($L_1$ valuation). An SCM $\mathcal{M} = \langle \mathbf{U}, \mathbf{V}, \mathcal{F}, P(\mathbf{U}) \rangle$ defines a joint probability distribution $P^{\mathcal{M}}(\mathbf{V})$ such that for each $\mathbf{Y} \subseteq \mathbf{V}$:

$$P^{\mathcal{M}}(\mathbf{y}) = \sum_{\mathbf{u}|\mathbf{Y}(\mathbf{u})=\mathbf{y}} P(\mathbf{u}) \tag{6}$$

Before we define $L_2$ evaluations, we need to understand interventional SCMs. Let $\mathcal{M}$ be an SCM and $\mathbf{x}$ be an assignment to $\mathbf{X} \subseteq \mathbf{V}$. Then the interventional SCM $\mathcal{M}_{\mathbf{x}}$ is the 4-tuple $\langle \mathbf{U}, \mathbf{V}, \mathcal{F}_{\mathbf{x}}, P(\mathbf{U}) \rangle$, where $\mathcal{F}_{\mathbf{x}} = \{f_i : V_i \notin \mathbf{X}\} \cup \{\mathbf{X} \leftarrow \mathbf{x}\}$. This operation is also known as the $do(\mathbf{x})$ operation.

**Definition A.4** ($L_2$ valuation). An SCM $\mathcal{M} = \langle \mathbf{U}, \mathbf{V}, \mathcal{F}, P(\mathbf{U}) \rangle$ induces a family a joint probability distributions over $\mathbf{V}$, one for each intervention $\mathbf{x}$. For each $\mathbf{Y} \subseteq \mathbf{X}$,

$$P^{\mathcal{M}}(\mathbf{y}_{\mathbf{x}}) = \sum_{\mathbf{u}|\mathbf{Y}_{\mathbf{x}}(\mathbf{u})=\mathbf{y}} P(\mathbf{u}) \tag{7}$$

where $\mathbf{Y}_{\mathbf{x}}(\mathbf{u}) = \mathbf{Y}_{\mathcal{M}_{\mathbf{x}}}(\mathbf{u})$

Such an operation, where the values of random variables $\mathbf{X}$ are set to constant values $\mathbf{x}$, is known as hard interventions. Conditional or stochastic interventions can be defined similarly Correa & Bareinboim (2020). Let $\sigma_{\mathbf{X}} = \{\sigma_X\}_{X \in \mathbf{X}}$ be the set of soft-interventions on the variables $X \in$

---

**Algorithm 3** `Ctf-RCT`: Counterfactual Randomized Controlled Trials in MAB

---

1: **Input:** domain of actions $\mathcal{D}(X)$, total number of trials $N \in \mathbb{N}$
2: **for** $t = 1, 2, \ldots$ **do**
3:     Perceive intended action $X^{(t)}$ and store it.
4:     **if** $t \leq N$ **then**
5:         Sample realized action

$$X'^{(t)} \sim \mathrm{Unif}\big(\mathcal{D}(X)\big).$$

6:     **else**
7:         Set

$$X'^{(t)} = \arg\max_x \widehat{\mathbb{E}}^{(N)}\big[Y_{X\leftarrow x} \mid X = X^{(t)}\big].$$

8:     **end if**
9:     Perform $\mathrm{do}\big(X'^{(t)}\big)$ and receive reward $Y^{(t)}$.
10: **end for**

---

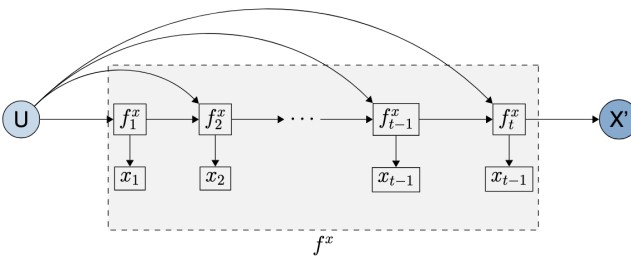

Figure 6: Illustration of decision flow $f_X$

**X**. Given $\sigma_\mathbf{X}$, the new model $M_{\sigma_\mathbf{X}}$ is defined as $\langle \mathbf{V}, \mathbf{U} \cup \mathbf{U}'_\mathbf{X}, \mathcal{F}', P(\mathbf{U} \cup \mathbf{U}'_\mathbf{X})\rangle$, where $\mathbf{U}'_\mathbf{X} = \{\mathbf{U}'_X\}_{X\in\mathbf{X}}$ and $\mathcal{F}' = \{\mathcal{F} \setminus \{f_X\}_{X\in\mathbf{X}}\} \cup \{f'_X\}_{X\in\mathbf{X}}$. The distribution $P(\mathbf{V}; \sigma_\mathbf{X})$ can then be computed as $P(\mathbf{V})$ in $M_{\sigma_\mathbf{X}}$.

Next, we move on to the $L_3$ distributions, where we ask questions of the form "Given that the patient died without the treatment, would they be alive if they were given the treatment?". The first thing to note here, is that this query is not the same as treatment effect, that is $E[Y_{x=1}] - E[Y_{x=0}]$, where we are taking the difference between the average effects of giving the treatment and not giving the treatment. On the other hand, in the counterfactual question, we are asking the question if it would have helped for the same individual. Now, the difficulty of this problem, lies in the fact, that the patient was already denied treatment and died, and it is not practical to go back in time and give them the treatment. Mathematically, if $Y$ is the variable that denotes whether is the patient is alive and $X$ be the variable that the patient was given the treatment, we can write the above question as $P(Y_{x=1} = 1 \mid X = 0, Y = 0)$. Now, we provide a formal definition on how to compute counterfactual queries, given an SCM.

**Definition A.5** ($L_3$ valuation). An SCM $\mathcal{M} = \langle \mathbf{U}, \mathbf{V}, \mathcal{F}, P(\mathbf{U})\rangle$ induces a family of joint distributions over counterfactual events $\mathbf{y_x}, \ldots \mathbf{z_w}$ for $\mathbf{Y}, \mathbf{Z}, \ldots, \mathbf{W}, \mathbf{X} \in \mathbf{V}$:

$$P^\mathcal{M}(\mathbf{y_x}, \ldots \mathbf{z_w}) = \sum_{\mathbf{u}|\mathbf{Y_x}(\mathbf{u})=\mathbf{y},\ldots\mathbf{Z_w}(\mathbf{u})=\mathbf{z}} P(\mathbf{u}) \tag{8}$$

The collection of observational ($L_1$), interventional ($L_2$) and counterfactual ($L_3$) are together called the PCH.

## A.2 COUNTERFACTUAL RANDOMIZATION

In practice, interacting through $L_3$ or counterfactual layer of the Pearl Causal Hierarchy can be extremely nontrivial. To this end, Bareinboim et al. (2015) introduces a novel form of randomization to interact through the Layer 3 of PCH. The challenge stems from the observation that agents may consider various alternatives during the deliberation process and change their opinion about the best

course of action. For example, if the agent initially considers $X = x$, and then reconsiders and changes to $X = x'$, is this counterfactual action where the natural intuition was $x$ and the performed action would be $x'$? What if the agent reconsiders their decision again and changes it to $x$; is the agent acting against their intuition? What is the intuition $x$ or $x'$. Counterfactual randomization Bareinboim et al. (2015; 2024) addresses this concern.

The main idea is that the agent may consider many options during the deliberation process, but only the final choice matters. Consider the deliberation process shown in Fig. 6: at time step $T = 1$, the agent intends to play $X = x_1$ but reconsiders, thinking it might be sub-optimal, and decides to switch to $X = x_2$ instead, where $x_1 \neq x_2$. As time passes, the agent may realize that $X = x_{t-1}$ was not ideal and switch to an alternative, $X = t$. Ultimately, the final decision defines the intuition of the agent, regardless of the path taken to reach it. In practice, the agent could also in this reasoning process forever without ever reaching a decision.

This challenge calls for novel counterfactual machinery to allow for the counterfactual interaction following layer 3. Bareinboim et al. (2015) introduces counterfactual randomization in which an agent is interrupted just before the execution of the choice, the choice being taken as the natural intuition and the final action executed based on this intuition and flip of a coin. Further, Bareinboim et al. (2024) also introduces `ctf-RCT`, where an intuition is observed and then an action is chosen at random for execution. This allows us to sample from counterfactual distributions of the form $P(Y_x \mid x') \, E[Y_x \mid x']$, where $Y$ is the outcome, $x$ is the intervened value and $x'$ is the intuition, measured just before the decision. The algorithm is shown in Alg. 3. For more details on this procedure in the single-agent setting, please refere to Bareinboim et al. (2024, Sec. 7)

### A.3 NORMAL FORM GAMES AND NASH EQUILIBRIUM

In many settings – from economics and political science to computer science and biology – multiple decision-makers interact strategically, each trying to achieve the best possible outcome for themselves. A *normal-form game* provides a compact way to model such one-shot interactions, and the concept of *Nash equilibrium* captures the idea of a stable outcome where no individual can benefit by unilaterally changing their choice. In this section of the appendix, we walk through these ideas step by step, illustrating them with the classical Prisoner's Dilemma and with the intent of contrasting this later on with other variations and approaches.

#### A.3.1 WHAT IS A NORMAL-FORM GAME?

Intuitively, a normal-form game asks:

> *"If each player picks an action simultaneously, how do their combined choices determine everyone's payoffs?"*

This question leads to the following definition of a game:

**Definition A.6** (Normal-Form Game). A finite $n$-player normal-form game is a tuple:

$$G = \langle N, \, A, \, u \rangle$$

where

- $N = \{1, 2, \ldots, n\}$ is the set of players.

- $A = A_1 \times A_2 \times \cdots \times A_n$, with each $A_i$ a finite set of actions available to player $i$. An element $a = (a_1, a_2, \ldots, a_n)$ is called an *action profile*.

- $u = (u_1, u_2, \ldots, u_n)$ is a collection of payoff functions, one per player:

$$u_i : A \longrightarrow \mathbb{R}, \quad a \mapsto u_i(a).$$

  Given a profile $a$, $u_i(a)$ tells us how much player $i$ "earns" (or how happy they are) under that combination of actions.

To summarize, in Normal Form Games:

- Each player $i$ simultaneously chooses an action $a_i \in A_i$.

- Once all choices $a = (a_1, \ldots, a_n)$ are made, each player $i$ receives payoff $u_i(a)$.

Despite their simplicity, Normal-Form Games are extremely powerful in their expressive power and many other richer representations, such as Extensive Form and Bayesian Games, can be reduced to a Normal Form equivalent. Next, we look at how agents can and should behave in a normal form games.

### A.3.2 MIXED STRATEGIES AND BEST RESPONSES

Rather than committing to a single action, players may *randomize* over their options. A *mixed strategy* for player $i$ is simply a probability distribution over $A_i$. Denote by

$$S_i = \Delta(A_i)$$

the set of all such distributions, and by $S = S_1 \times \cdots \times S_n$ the collection of all players' mixed strategies.

Given that the other players use some mixed-strategy profile $s_{-i} \in S_{-i}$, player $i$ will choose a distribution $s_i \in S_i$ to maximize their expected payoff

$$u_i(s_i, s_{-i}) = \sum_{a \in A} \big[ s_i(a_i) \times s_{-i}(a_{-i}) \big]\, u_i(a).$$

**Definition A.7** (Best Response). A mixed strategy $s_i^* \in S_i$ is a *best response* to opponents' strategy profile $s_{-i}$ if

$$u_i\big(s_i^*, s_{-i}\big) \geq u_i\big(s_i, s_{-i}\big) \quad \text{for every } s_i \in S_i.$$

In other words, $s_i^*$ gives player $i$ the highest possible expected payoff, assuming the others stick to strategy $s_{-i}$.

The notion of best response will play a key role in understanding agent's behavior.

### A.3.3 NASH EQUILIBRIUM

A Nash equilibrium is a collection of strategies – one per player – such that each player's choice is a best response to everyone else's. No one can gain by deviating alone. Such a notion gives a concept of stability in a game, or a type of a solution.

**Definition A.8** (Nash Equilibrium). A mixed-strategy profile $s^* = (s_1^*, \ldots, s_n^*)$ is a *Nash equilibrium* if, for every player $i$,

$$u_i\big(s_i^*, s_{-i}^*\big) \geq u_i\big(s_i, s_{-i}^*\big) \quad \text{for all } s_i \in S_i.$$

### A.3.4 EXAMPLE: THE PRISONER'S DILEMMA

To see these definitions in action, consider the *Prisoner's Dilemma*, a two-player game where each must choose either to cooperate ($C$) or defect ($D$). The actions of the player $i$ can be written as:

$$A_i = \{C, D\}.$$

Their payoffs are given by the following matrix (first entry is player 1's payoff, second is player 2's):

| P1 \ P2 | C | D |
|---|---|---|
| C | $-1, -1$ | $-7, -0.5$ |
| D | $-0.5, -7$ | $-1.9, -1.9$ |

Let's walk through each prisoner's incentives:

- If player 2 cooperates ($C$), then player 1's payoff is

$$u_1(C, C) = -1 \quad \text{vs.} \quad u_1(D, C) = -0.5.$$

player 1 is better off defecting ($D$), since $u_1(D, C) > u_1(C, C)$.

| | Convict 2 | $X_2 = 0$ | $X_2 = 1$ |
|---|---|---|---|
| Convict 1 | | | |
| $X_1 = 0$ | | $-1, -1$ | $-7, -0.5$ |
| $X_1 = 1$ | | $-0.5, -7$ | $-1.9, -1.9$ |

(a) Causal graph for Markovian Prisoner's Dilemma     (b) Payoff matrix for Prisoner's Dilemma

Figure 7: Representation of the Markovian Prisoner's Dilemma: (a) causal graph and (b) corresponding payoff matrix.

- If player 2 defects ($D$), then player 1's payoff is

$$u_1(C, D) = -7 \quad \text{vs.} \quad u_1(D, D) = -1.9.$$

  Again, player 1 prefers to defect ($D$), since $u_1(D, D) > u_1(C, D)$.

- Hence, it is better for player 1 to defect $D$, irrespective of what player 2 does. By symmetry, player 2 likewise always prefers $D$ whatever player 1 does.

Thus, each player's unique *best response* to the other is to defect. When both play their best responses, we reach the profile, $(D, D)$ which is the game's *Nash equilibrium*. Ironically, although $(C, C)$ would yield $(-1, -1)$ total payoff (mutual cooperation), rational self-interest drives both to $(D, D)$, giving only $-1.9 - 1.9 = 3.8$.

### A.4    PARADOX OF RATIONALITY

It has been observed throughout economics and behavioral game theory literature that irrationality can result in better outcomes than rational choices. One such example is the prisoner's dilemma. Both cooperating would be an irrational choice, but it results in a better payoff compared to fully rational players both of whom would choose to confess. Such irrational co-operations have also been observed in practice Colman (2003). There has been several attempts in order to explain such irrationalities observed in human decision-making either through different models of bounded rationality, such as payoff transformations Tversky & Kahneman (1981); Kahneman & Tversky (1984; 2013) or through alternate forms of reasoning Colman (2003).

Consider the example of the Travelers' Dilemma Basu (1994), where 2 travelers are asked to write the price of their lost item between $2-100. One with the lower value receives the lower value + $2 and one with the higher value receives lower value - $2. If an agent just tries to maximize their own reward and do not reason over others, both of them will write $100 and receive that. Now, if they do one step of reasoning, they will think "If I write $99 and my opponent writes $100 then, I will get $101 and my opponent $97". Hence, both will write $99 and get $99. The amount will decrease with more levels of reasoning. Irrational players again get higher payoffs than rational agents.

Basu (1994) states that different thought processes lay behind different types of choices that people made playing a version of Traveler's Dilemma with the options ranging from 180 to 300 (pie chart): a spontaneous emotional response (choosing 300), a strategically reasoned choice (295–299) or a random one (181–294). Players making the formal rational choice (180) might have deduced it or known about it in advance. As expected, people making "spontaneous" or "random" selections took the least time to choose (as seen in experiments).

### A.4.1    CAUSAL GAME THEORY AND PARADOX OF RATIONALITY

Our proposed framework can both model and explain this gap between theory and practice. First, we consider the modeling part through the example of Prisoner's Dilemma, which we will also call the Markovian Prisoners Dilemma. The causal graph for the Markovian PD is shown in Fig. 7a. Let $X_1$ and $X_2$ be the action variables and their values 0 and 1 correspond to cooperating ($C$) and defect ($D$) respectively, and let their natural probability of cooperating be $P(X_i = 0) = 0.9$ for $i \in \{1, 2\}$. If someone is simply collecting observational data, it may happen that the agents are simply playing $L_1$ and hence the corresponding payoff is higher. Thus our modeling of games can model the irrational tendencies of the agents involved.

Next, comes the explaining part. Consider the scenario $M_1$ in Ex. 1.1, and the optimal equilibrium action, which is both the players playing $L_1$. Note that, this is infact the best choice from the perspective of a player, who knows how to act in $L_1, L_2$ or $L_3$. However, from an external observers point of view, these players are playing suboptimally, that is, playing $C$ with probability $0.6$ and $D$ otherwise. From an external's observers' point of view, if the agents performed RCT, they would have a payoff as shown in Fig. 1b (bottom table), according to which playing $D$ is the optimal strategy with a payoff $(-1.9, -1.9)$. However, the agents, playing seemingly irrationally somehow get a payoff of $(0, 0)$, creating a paradox in the mind of the external experiment designer.

### A.5 $L_1, L_2$ AND $L_3$ ACTIONS IN SINGLE-AGENT SYSTEMS: THE GREEDY CASINO

The following examples illustrates the limitations of traditional decision-making and how an agent can interact with the system through the three layers of PCH as first introduced by Bareinboim et al. (2015). Consider a casino introducing two new slot machines, denoted $0$ and $1$. Gamblers choose machines according to two unobserved binary factors: their level of inebriation ($D \in \{0, 1\}$) and whether a machine is blinking ($B \in \{0, 1\}$). Although these factors are hidden from the agent, they influence natural behavior through the rule $X = D \oplus B$, determining the arm $X \in \{0, 1\}$ a gambler is predisposed to choose.

The casino exploits this behavioral pattern by designing reactive slot machines that adjust payouts based on these hidden variables. While ensuring that payout rates meet a government-mandated minimum of 30% when players are assigned arms randomly (e.g., RCT during inspection), the machines covertly reduce payout rates for players who follow their natural inclinations. The effective payouts are given in Table 1.

|       | $D = 0$ |       | $D = 1$ |       |
|-------|---------|-------|---------|-------|
|       | $B = 0$ | $B = 1$ | $B = 0$ | $B = 1$ |
| $X = 0$ | **0.10** | 0.50 | 0.40 | **0.20** |
| $X = 1$ | 0.50 | **0.10** | **0.20** | 0.40 |

Table 1: Reactive slot machine payouts: bolded entries indicate natural arm choices under the rule $X = D \oplus B$.

Note, that while players are following their natural choice, the payoff is given by

$$E[Y] = \sum_{b,d} y \cdot P(y \mid X = d \oplus b, b, d) = 0.1 \tag{9}$$

On the other hand, if the inspectors do an RCT, the payoff is given by

$$E[Y \mid do(X = x)] = \sum_{b,d} y \cdot P(y \mid X = x, b, d) = 0.3 \tag{10}$$

for any $x \in \{0, 1\}$.

However, if the agents are following opposite of their natural intuition, then the payoff will be given by

$$E[Y_{X=0} \mid X = 1]P(X = 1) + E[Y_{X=1} \mid X = 0]P(X = 0) = 0.45 \tag{11}$$

which is significantly higher than the other strategies. In fact, we can now make the following observation:

$$E[Y_x \mid x'] > E[Y_{x'} \mid x'] = E[Y \mid x'] \tag{12}$$

for any $x \neq x'$. The first term corresponds to the scenario when the gambler wants to play $x'$, but then just before execution they choose $x$. The payoff of such a strategy is higher than the payoff of just playing their intuition. The term where they simply choose a single machine and play (as in RCT), $E[Y'_x]$ lies between these two strategies. In general $L_3$ strategies will always outperform $L_1$ and $L_2$ strategies, since both can be expressed as $L_3$ strategies.

In fact for single agent systems, it is always better for the agents to follow $L_3$ policy space, as $L_3$ space subsumes $L_1$ and $L_2$ policies.

$$\max_\pi \sum_x E[Y_{X=\pi(x)} \mid x] P(x) \geq \max\{\max_a E[Y_{X=a}], E[Y]\} \tag{13}$$

For more details, please refer Bareinboim et al. (2015; 2024).

### A.6 GRAPHICAL MODELS AND GAME THEORY

Several works have studied game theory from a graphical models perspective. The main emphasis has been on the computational advantages related to learning equilibria through probabilistic reasoning and corresponding optimization tools Koller & Milch (2003); Kearns et al. (2001). This is a part of the growing and important literature known as algorithmic game theory Roughgarden (2010). Our approach addresses key gaps in existing models, particularly concerning the assumption of Markovianity, issues of irrationality, and multi-agent interactions.

Specifically, Kearns et al. (2001) introduced *graphical games* to leverage graph structures for modeling interactions among players, making equilibrium computation more efficient when compared to standard Normal Form Games. Furthermore, Koller & Milch (2003) extended influence diagrams Howard et al. (1990); Lauritzen & Nilsson (2001) to multi-agent settings, where decision nodes represent strategies, and probabilistic dependencies simplify equilibrium computations. Their framework was called Multi-Agent Influence Diagrams (MAIDs). The main goal of these works was connecting graphical models and game theory, and where somewhat silent with respect to how this relate to causality, including interventions and counterfactual reasoning.

The Structural Causal Influence Model by Everitt et al. (2021) connects causality with the influence diagrams literature Howard et al. (1990); Lauritzen & Nilsson (2001). They study certain notions found in this traditional literature, including value of information, value of control, among others. Their setting focuses on single-agent settings, whereas this paper considers multi-agent interactions, including more equilibrium analysis in scenarios where agents compete in a strategic manner. They also did not consider unobserved confounding, which is one of the key challenges in typical causal settings. Another form of causal games is proposed by Gonzalez-Soto et al. (2019), which again focuses on actions as interventions and ignores the other layers of operations by a player.

Hammond et al. (2021) extends Koller & Milch's MAIDs by introducing the concept of MAID subgames and proposing equilibrium refinements such as subgame perfect and trembling hand perfect equilibria. The authors establish equivalence results between MAIDs and Extensive Form Games (EFGs), highlighting the computational advantages of MAIDs in representing and solving certain classes of games. Still, despite its power, this work does not explore causal implications or counterfactual strategies, which are central to our framework. Our model explicitly integrates these aspects for deeper insights into strategic decision-making and the meaning of rationality.

Unlike the Structural Causal Games framework in Hammond et al. (2023), which assumes Markovian dynamics, our model handles non-Markovian influences, including unobserved confounding that impact both actions and payoffs. We note that the assumptions required to ascertain Markovianity are inapplicable in our setting, since one of our main goals is to account for irrational behavior – where the agent acts without knowing why. In a Markovian setting, the agent knows the reasons for acting in a particular way. In fact, we model irrationality through the notion of counterfactuals and extend equilibrium concepts beyond purely rational agents, as prescribed by Nash's framework. A detailed comparison with this work is provided in the next section.

The approach proposed by Chan et al. (2021) embeds irrationality in the Bellman equation under a Markovian assumption in a novel way. Our model, however, allows for general irrationality without specifying any functional constraints, which is necessary in a non-Markovian setting. The assumptions required to ascertain Markovianity are inapplicable in our setting, since one of our main goals is to account for irrational behavior – where the agent acts without knowing their reasons. Furthermore, while their focus is on a single-agent environment, ours is on multi-agent, strategic settings.

By bridging these gaps, our model provides a unified view of rational and irrational behaviors through a causal lens and rooted in first principles. It also extends graphical game-theoretic models to multi-agent systems, contributing to a more comprehensive understanding of equilibrium dynam-

ics and rationality. Notably, while our work falls within the realm of causality, it is not primarily focused on its graphical aspects, as evident throughout the main body of the paper. As mentioned earlier, the central issue addressed here concerns the most fundamental decision-making setting and how counterfactual reasoning (and counterfactual randomization Bareinboim et al. (2015)) can be leveraged to model and reconcile both irrational and rational behaviors, ultimately resolving the rationality paradox. We believe that the foundational understanding developed in this pervasive setting can be generalized to more complex games, where a graphical model and a more fine-grained structure could play a role, including for computational purposes.

## A.7 Notes on Hammond et al. (2023)

We note that Hammond et al. (2023) also claim to unify causal modeling and game theory through the formalism of causal games and structural causal games (SCGs). In this section, we provide a critical examination of this claim and demonstrate that their approach fails to capture several essential aspects of causal modeling that are fundamental to understanding how agents reason and act in complex systems. Specifically, we identify four key limitations in their formalism: (1) the absence of the causal hierarchy and associated distributions, (2) the neglect of unobserved confounding, (3) the inability to represent $L_1$ actions even when extended with default functions, and (4) a flawed approach to counterfactual evaluation that misinterprets key semantic and identification issues. Each of these points is discussed in detail below.

**1. Absence of the Causal Hierarchy and their distributions:** One of the most basic features of causal modeling is the presence of different probability distributions induced by the collection of causal mechanisms, which is organized as three qualitatively different probability distributions – observational, interventional, and counterfactual Pearl & Mackenzie (2018); Bareinboim et al. (2022), and which are also known as the Pearl Causal Hierarchy (PCH). These three levels of distributions separate causal models from previously used graphical models such as Bayesian networks, and are considered a novel landmark in evaluation, estimation, representation, and decision-making in complex environments. As a consequence, an agent can interact with the system in three different ways corresponding to the three layers of PCH policies: $L_1, L_2,$ and $L_3$. However, Hammond et al. (2023) collapse this fundamental hierarchy into a single layer by treating all agent actions as (hard and soft) interventions, disregarding the observational ($L_1$) and counterfactual ($L_3$) levels of reasoning completely. Recall the key definition of games introduced in this work (Hammond et al., 2023, Def. 22):

**Definition A.9** (SCG). A (Markovian) SCG $M = (G, \theta)$ is a causal game over the exogenous and endogenous variables $\mathbf{E} \cup \mathbf{V}$ such that any deterministic parameterization of the decision variables of CPD $\pi$, the induced model with join distribution $P^\pi(V, E)$ is an SCM.

The authors explain that:

> "An SCG can be seen as an SCM without parameters for the decision variables. Given a policy $\pi$, we recover an SCM, as we explain in more detail below."

As a result, an SCG itself does not define a natural distribution ($L_1$), because the decision variables, say $X$ do not have a natural mechanism $f_X$ and are only determined by the agents or the policy. This precludes the possibility of modeling agents that interact in a $L_1$ or $L_3$ manner, significantly restricting the expressivity of SCGs in modeling how real-world agents reason. Still, in the formalism introduced in this paper, there is a natural distribution of the decision variables (Layer 1 in the PCH), hard/soft interventions (Layer 2), and counterfactual actions (Layer 3). Such fundamental features could not be captured by the models proposed in Hammond et al. (2023), which is illustrated in the following example.

**Example A.10** (Markovian Prisoner's Dilemma). Two thieves are suspected of a crime and are captured. Unfortunately, there is not enough evidence to convict them. They can now cooperate ($X = 0$) or defect ($X = 1$). The payoffs for the actions of the convicts are shown in Fig. 7b, where the numbers can be interpreted as the years they have to serve in prison. The Nash equilibrium of this game is when both players defect and the payoffs are $(-2, -2)$, where both players have no incentive to cooperate. The causal diagram for such a scenario is shown in Fig. 7a.

Consider the following scenarios – in the first one, let us call it $M_1$, the prisoners are more loyal and their spontaneous instinct is to cooperate ($X = 0$) with a probability of 0.9 (disregarding their

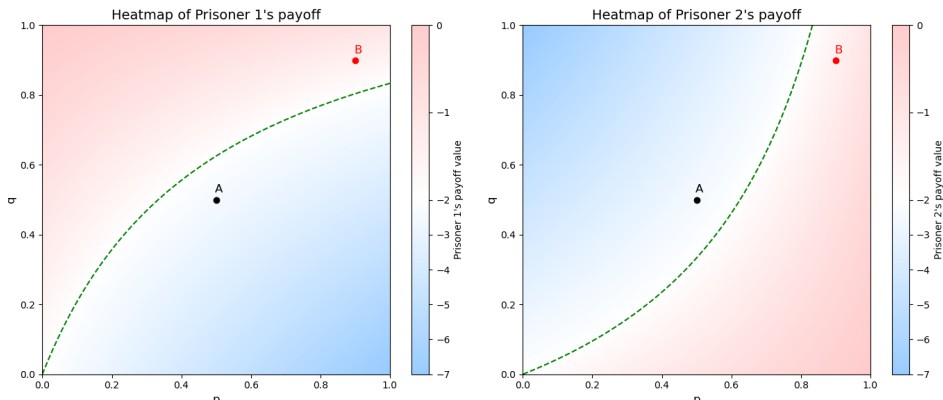

Figure 8: Payoffs of the two players as a function of their lying probability. $A$ and $B$ denote the payoffs when the agents follow their natural instincts in scenario $M_1$ and $M_2$, respectively.

utility), and in the second scenario, called $M_2$, the instinct is to cooperate ($X = 0$) with a probability of 0.5 and confess otherwise. Now, if the agents follow their natural instincts, their payoffs in the first scenario are

$$\mu_{L_1}^1 = \sum_{x_1, x_2} \mathbf{Y} \cdot P(X_1 = x_1) P(X_2 = x_2) P(\mathbf{Y} \mid x_1, x_2) = (-1.46, -1.46)$$

and in the second scenario is

$$\mu_{L_1}^1 = \sum_{x_1, x_2} \mathbf{Y} \cdot P(X_1 = x_1) P(X_2 = x_2) P(\mathbf{Y} \mid x_1, x_2) = (-2.5, -2.5)$$

If $\mu_{\text{NE}}$ is the NE payoff, we can see that

$$\mu_{L_1}^1 > \mu_{\text{NE}} > \mu_{L_1}^2 \tag{14}$$

Now, these natural distributions cannot be represented or modeled by an SCG, where the $X_1$ and $X_2$ are determined by interventions. Hence, agents cannot act in $L_1$ and SCGs cannot capture the subtlety in Eq. 14. In fact, both $M_1$ and $M_2$ result in the same SCG in Fig. 7a and mechanized SCG in Fig. 9. In fact $M_1$ and $M_2$ are only two instances of infinitely many more scenarios that can happen. Fig. 8 shows how the players' $L_1$ payoffs change with their probability of cooperating. The scenarios $M_1$ and $M_2$ are marked as $A$ and $B$ in the plots. The green line denotes $L_1$ payoffs equal to the NE payoff. Also, since there is no concept of natural actions, $L_3$ policies also do not exist in SCGs. For example, in $M_2$ if both the prisoners decide to act against their natural instinct, then the payoffs are $\mu_{L_3}^2 = (-1.46, -1.46)$ and hence $\mu_{L_3}^2 > \mu_{\text{NE}}$.

**2. Unobserved Confounders:** One of the major challenges in real-world settings that causal inference is concerned with is the existence of unobserved confounding, variables that cannot be measured but influence both decisions and outcomes. The classic saying that "causation is not association" comes precisely because of the existence of such confounders. However, Hammond et al. (2023, Sec. 2.1) does not take unobserved confounders into account.

> "In this paper, we make the simplifying assumption that all SCMs are Markovian, meaning that each variable $V$ has exactly one exogenous parent $E_V$ and the exogenous variables are independent."

The problem becomes even more fundamental in the context of causal game theory, as highlighted in the Causal Prisoner Dilemma (Ex. 1.1). In words, both $M_1$ and $M_2$ imply the same SCG and mechanized-MAID (Fig. 9), but entail entirely different causal analysis and decision-making strategies. For example, in $M_1$, following the $L_1$ strategies is the equilibrium strategy, and in $M_2$ following the $L_3$ strategies are the equilibrium strategies, and in both cases the equilibrium is better than the equilibrium payoffs of $L_2$. However, this observation is not an idiosyncrasy but rather part of a broader phenomenon, as highlighted by the following proposition.

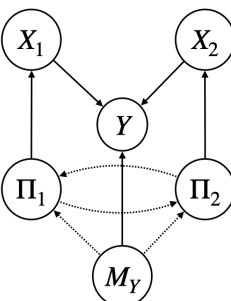

Figure 9: m-MAID is same for both $M_1$ and $M_2$ in Ex. 1.1

**Theorem A.11** (CNFG representation of SCG). *Given an SCG corresponding to a Normal Form Game, there exist two CNFGs $\mathcal{C}_1$ and $\mathcal{C}_2$, with equilibrium payoffs $\mu_1$ and $\mu_2$ under $L_1$ and $L_2$ actions and Nash Equilibrium payoffs $\mu_{NE}$, such that*

$$\mu_1 < \mu_{NE} < \mu_2 \tag{15}$$

*Proof.* This result follows from Thm. 2.11. Recall Def. 22 from Hammond et al. (2023), and Ex. 1.1 of this work. Hence, if we follow the proposed construction, we see there is a bijection between a normal form game and its SCG representation, which implies that given a normal form game there is a single SCG which represents the same game as the normal form game. However, Thm. 2.11 implies the existence of two CNFGs satisfying Eq. 15. Hence, CNFG represents a larger class of models than SCGs for normal form games. $\square$

Formally, CNFGs is a strictly larger class than normal form games represented as SCG.

**3. Why can't an SCG with default actions represent $L_1$ actions?**

One may surmise that a Strategic Causal Game (SCG) could be extended to include a default action to represent an $L_1$ action. However, this would face two fundamental challenges:

1. **Defining the Default Action:** In CNFGs, $L_1$ actions are determined by nature (the SCM) through mechanisms that are entirely unknown to the agent. For instance, in Example 1.1, suppose we attempt to proxy an $L_1$ action in an SCG by defining a default function such as $X = U \oplus R$. But why should this specific function be chosen? It could just as well be $U \cdot R, U \vee R$, or any arbitrary combination of $U$ and $R$. Since the agent has no knowledge of these variables or the functional form governing them, they cannot meaningfully prefer one default over another. This renders the notion of a default action indeterminate from the agent's perspective.

2. **Dependence on Unobserved Variables:** A default action that depends on unobserved variables (like $U$ or $R$) inherently contradicts the assumption that these variables are unobservable to the agent. If the agent is able to use these variables as inputs in its decision-making process, then, by definition, they are no longer unobserved. This undermines the epistemic foundations on which the model is built. Put differently, this denies the possibility of unobserved confounding, a common challenge in causal inference.

These issues highlight a deeper point: SCGs and CNFGs represent fundamentally different models of agent-environment interactions. Attempting to equip SCGs with mechanisms to mimic CNFG behavior would ultimately collapse the SCG framework into that of CNFG.

**4. Counterfactual Evaluation:** Next, we look at the evaluation of counterfactual queries and the rationale behind computing such queries, as proposed in Hammond et al. (2023). In words, they want to answer "If we have evidence that the equilibrium $\pi$ was played in the actual world, how and to what extent should that inform us of the equilibrium $\pi'$ played in the counterfactual world where the values of some mechanism variables may have changed?" They claim to compute the quantity $P^{\pi'}(\mathbf{x}_I \mid z_\pi)$ through the following procedure:

- For every actual rational outcome $\pi \in \mathcal{R}(\mathcal{M} \mid z)$, update $P(\mathbf{u})$ to $P(\mathbf{u} \mid z)$ ('abduction')
- Apply the intervention $I$, on variables $\mathbf{Y}$, recomputing any rational responses to form $\pi'$ and adding new exogenous variables as required ('action')
- Return each marginal distribution $\int_{D_{\mathbf{u}}} P^{\pi'}(\mathbf{x} \mid \mathbf{u}')P(\mathbf{u}')d\mathbf{u}'$ in the modified model for each counterfactual rational outcome $\pi'$ ('prediction')

Note that even though this follows Pearl's algorithm in theory, it misses a fundamental practical point: neither $\mathbf{u}$ is observed, nor is $P(\mathbf{u})$ known, which makes the above evaluation impossible in most cases (Pearl, 2009). Formally, this procedure provides clear semantics for counterfactuals (Bareinboim et al., 2022, Sec. 1.2), but it does not immediately imply their identification. Methods to overcome this impossibility of directly evaluating counterfactual quantities are known in the literature (see, e.g., (Correa et al., 2021; Raghavan & Bareinboim, 2025a)). For example, one counterfactual quantity that Hammond et al. (2023) would be interested in, within our context from Example1.1, is

$$P(Y_{X_1=1,X_2=0} = y \mid Y_{X_1=0,X_2=0} = y'), \tag{16}$$

which is known not to be identifiable from observational or interventional data without further assumptions. We note that their counterfactual evaluations do not correspond to the counterfactual actions studied in this paper. In contrast, we compute the quantity

$$\sum_x P(Y_{X_1=x',X_2=0} \mid X_1 = x)P(X_1 = x), \tag{17}$$

which SCGs cannot represent, since they do not define a natural distribution. Moreover, this quantity is counterfactually realizable and can be used in practice through counterfactual randomization Raghavan & Bareinboim (2025a).

# B PROOFS

## B.1 PROOF OF THEOREM 2.11

Consider a normal form game $\mathcal{G}$ with the action space $A = A_1 \times \ldots \times A_n$ and the utility function $r = (r_1, \ldots, r_n)$. Assume all the utilities are finite. Suppose $s^*$ is the NE strategy and $\mu_{NE}$ is the NE payoff.

Next, we will construct an SCM that induces the same Normal Form Game under $L_2$ actions as follows:

- $\mathbf{U} = \{U_1, \ldots, U_n\}$, $D_{U_i} = A_i$ for all $i \in [n]$
- $\mathbf{X} = \{X_1, \ldots, X_n\}$, $D_{X_i} = A_i$ for all $i \in [n]$
- $X_i = U_i$ for all $i \in [n]$
- $P(U_i = a_i^j) = s_i^*(a_i^j)$ where $s_i^*(a_i^j)$ is the probability of playing $a_i^j$ by agent $i$ in the NE strategy $s_i^*$.
- Now, we will define the observed variable the payoff of the $i$-th agent for $i \in [n]$. For $i \neq 1$, we define $Y_i(a, \mathbf{u}) = r_i(a)$, for all $a \in D_{\mathbf{X}}$. For $i = 1$, we have

$$Y_1(a, \mathbf{u}) = r_1(a) + M \cdot (\mathbb{1}\{U_1 = a_1\} - s_1^*(a_1)) \tag{18}$$

Now it is easy to see that for all $i \neq 1$,

$$\mathbb{E}[Y_i \mid do(a)] = \sum_{\mathbf{u}} Y_i(a, \mathbf{u})P(\mathbf{u}) = r_i(a) \sum_{\mathbf{u}} P(\mathbf{u}) = r_i(a) \tag{19}$$

For $i = 1$ and for all $a \in D_{\mathbf{X}}$,

$$\mathbb{E}[Y_1 \mid do(a)] = \sum_{\mathbf{u}} Y_1(a, \mathbf{u})P(\mathbf{u}) \tag{20}$$

$$= \sum_{\mathbf{u}} \left(r_1(a) + \mathbb{1}\{U_1 = a_1\} \cdot M \cdot (|A_1| - 1) - \mathbb{1}\{U_1 \neq a_1\} \cdot M\right)P(\mathbf{u}) \tag{21}$$

$$= r_1(a) + M \cdot (P(U_1 = a_1) - s_1^*(a_1)) \tag{22}$$

$$= r_1(a) \tag{23}$$

Hence, for $L_2$ action space, the SCM induces the same normal form game $\mathcal{G}$.

For $\Gamma_1$, suppose $M$ is a significantly large positive number and for $\Gamma_2$, let $M$ be a significantly large negative number, then we have for agent 1, the $L_1$ payoff is higher in $\Gamma_1$ and lower in $\Gamma_2$ than the NE payoff; for all other agents the payoff is the same. Hence, it follows, that

$$\mu_2 < \mu_{\text{NE}} < \mu_1$$

$\square$

## B.2 Proof of Theorem 3.5

Let $\Gamma$ be a CNFG and consider the associated Layer Selection Game (LSG) $L_\Gamma$. By construction, $L_\Gamma$ is a finite normal form game, since each agent chooses a reasoning layer from a finite set, and the payoffs are well-defined as the NE payoffs of the induced subgames. By Nash's theorem, $L_\Gamma$ admits at least one mixed strategy Nash equilibrium. Let this equilibrium strategy profile be $s^* = (s_1^*, \ldots, s_n^*)$.

For each agent $i$, let

$$A_i^* = \text{supp}(s_i^*) \tag{24}$$

the set of action spaces played with positive probability in $s_i^*$. Define the restricted action space

$$A^* = A_1^* \times \cdots \times A_n^* \tag{25}$$

The PCH projection of $\Gamma$ with action space $A^*$ is then a subgame of $\Gamma$. This subgame can be represented in normal form, where each player's strategy space is finite. Hence, by Nash's theorem again, this subgame admits at least one Nash equilibrium.

The resulting equilibrium constitutes a Causal Nash Equilibrium (CNE) of the original CNFG $\Gamma$. Therefore, a CNE exists for every CNFG. Moreover, just as normal form games may have multiple Nash equilibria, CNFGs may admit multiple CNEs.

## B.3 Proof of Theorem 3.6

First note that $\mu^* = NE(\Gamma(A^*))$. Suppose an agent is able to change the action space from $A_i^*$ to $A_i'$ and improve their payoff. However, if that was true, then $NE(\Gamma(A_i', A_{-i}^*)) > NE(\Gamma(A_i^*, A_{-i}^*))$, which implies in the PCH-LSG $L_\Gamma$, agent $i$ would be able to improve the payoff moving from $A_i^*$ to $A_i'$. However, by our assumption $A^*$ is the pure strategy NE of $L_\Gamma$, hence no such deviations are incentivised - a contradiction. Hence $\mu^* \geq NE(\Gamma(A_i', A_{-i}^*)$ for all

Let $\Gamma$ be a CNFG and let $L_\Gamma$ be its layer–selection game. Assume $L_\Gamma$ admits a pure–strategy Nash equilibrium, denoted $A^* = (A_1^*, \ldots, A_n^*)$, where $A_i^*$ is the action–space (PCH layer set) chosen by player $i$. Let $\mu^* = \text{NE}(\Gamma(A^*))$ be the Nash–equilibrium payoff vector of the PCH projection of $\Gamma$ in which the action spaces are restricted to $A^*$.

Fix a player $i$ and any alternative admissible action–space $A_i'$ for player $i$, $A_i' \in \{A_i^1, A_i^2, A_i^1 \cup A_i^2, A_i^3\}$. Suppose, that deviating to $A_i'$ would strictly improve player $i$'s payoff, that is,

$$\text{NE}_i\big(\Gamma(A_i', A_{-i}^*)\big) \; > \; \text{NE}_i\big(\Gamma(A^*)\big) = \mu_i^*.$$

By the definition of $L_\Gamma$, the payoff to player $i$ from choosing an action–space is precisely the equilibrium payoff of the corresponding PCH–projected game. Hence, holding $A_{-i}^*$ fixed, player $i$ would obtain a strictly higher payoff in $L_\Gamma$ by switching from $A_i^*$ to $A_i'$. This contradicts the fact that $A^*$ is a Nash equilibrium of $L_\Gamma$.

Therefore, for every player $i$ and every admissible unilateral deviation $A_i'$,

$$\mu_i^* \; \geq \; \text{NE}_i\big(\Gamma(A_i', A_{-i}^*)\big).$$

Hence, the theorem follows.

## B.4 Proof of Theorem 4.1

First, we will show that the payoff matrix learned is a permutation of the true payoff matrix, and then find out why $L_2$ or $L_3$ payoffs will be properly learned. First note that, since the mixture is identifiable, we recover the $D_{X_2} = k$ distributions, where each of them corresponds to a value of $X_2'$. However the deduced values $\{\hat{x}_2^1, \ldots, \hat{x}_2^k\}$ are arranged in decreasing order of distribution, that is

$$p(\hat{x}_2^1) > \ldots > p(\hat{x}_2^n)$$

In reality, the original values of $X_2'$ may not be so well arranged and hence $\{\hat{x}_2^1, \ldots, \hat{x}_2^k\} = h(\{x_2^1, \ldots, x_2^k\})$ where $h$ is a permutation function.

Now, $L_3$ action space consists of all the functions from natural intuition $X_2'$ to $X_2$. Hence the values of $X_2'$ are essentially irrelevant and we can learn the whole table upto a permutation of the action of the second player. Since NE of Player 1 and the NE payoff remains same even with the permutation of the action space, we have that $NE(\Gamma(A_1^3, A_1^3))$ will be properly learned.

Now, $L_2$ action spaces are constant functions and remain invariant to permutations of $X_2'$. Hence, in a similar manner $NE(\Gamma(\mathcal{A}_1^2, \mathcal{A}_2^2))$ will be correctly learned, as will $NE(\Gamma(\mathcal{A}_1^3, \mathcal{A}_2^2))$ and $NE(\Gamma(\mathcal{A}_1^2, \mathcal{A}_2^3))$, and so on. By our assumption, the NE strategy of the PCH-LSG for the other agent spans over $\mathcal{A}_2^2$ and $\mathcal{A}_2^3$. Hence, the NE strategy of PCH-LSG lies on the space $\{\mathcal{A}_1^1, \mathcal{A}_1^2, \mathcal{A}_1^1 \cup \mathcal{A}_1^2, \mathcal{A}_1^3\} \times \{\mathcal{A}_2^2, \mathcal{A}_2^3\}$. Since, we are able to learn NE corresponding to each of these policies, we can correctly identify the CNE strategy.

Let $\Gamma = \langle M, (\mathcal{A}_1^3, \mathcal{A}_2^3), R \rangle$ be a two–player CNFG and let the data be generated by Ctf-RCT. For fixed $(x_1', x_1, x_2)$ the outcome distribution of $Y$ is a finite mixture whose components correspond to the latent values of the opponent's intuition $X_2'$; by the identifiability assumption of the mixture, Algorithm 2 recovers the $k = |D_{X_2'}|$ component means and weights, but only up to a permutation of the component index. Formally, there exists a permutation $\pi$ of $\{1, \ldots, k\}$ such that the learned pairs $\left(\hat{p}_j(x_1', x_1, x_2), \hat{R}_j(x_1', x_1, x_2)\right)$ satisfy

$$\hat{p}_j(x_1', x_1, x_2) = p_{\pi(j)}(x_1', x_1, x_2), \qquad \hat{R}_j(x_1', x_1, x_2) = R_{\pi(j)}(x_1', x_1, x_2), \qquad (26)$$

for all $j = 1, \ldots, k$ and all $(x_1', x_1, x_2)$. Hence the payoff table that Algorithm 2 constructs for the $L_3$ action game is isomorphic to the true payoff table via a relabeling of the columns indexed by the opponent's intuition values.

Consider first the projection $\Gamma(A_1^3, A_2^3)$. Player 1's $L_3$ action space is $F_1 = \{f : D_{X_1'} \to D_{X_1}\}$ and Player 2's $L_3$ action space is $F_2 = \{g : \{1, \ldots, k\} \to D_{X_2}\}$. Because the learned table differs from the true table only by the permutation $\pi$ of the intuition labels, replacing each $g \in F_2$ by $g \circ \pi$ yields a bijection between the learned and true games that preserves payoffs. Therefore the two normal form games $\Gamma(A_1^3, A_2^3)$ (true) and $\widehat{\Gamma}(A_1^3, A_2^3)$ (learned) are strategically equivalent; in particular, they have the same set of Nash equilibria and the same equilibrium payoffs. Thus Algorithm 2 learns $NE(\Gamma(A_1^3, A_2^3))$ correctly.

Next consider projections that use $L_2$ actions for Player 2. An $L_2$ action for Player 2 is a constant function $c : \{1, \ldots, k\} \to D_{X_2}$. Such constants are invariant under any permutation of the intuition labels; hence the learned payoff table for $\Gamma(A_1^3, A_2^2)$ (and, symmetrically, for $\Gamma(A_1^2, A_2^2)$ and $\Gamma(A_1^2, A_2^3)$) coincides with the true one. Consequently the corresponding Nash equilibrium payoffs are learned correctly.

By the hypothesis of the theorem, in the layer selection game $L_\Gamma$ the equilibrium action space of Player 2 is $A_2 \in \{A_2^2, A_2^3\}$. From the previous paragraphs, for every action space $A_1$ of Player 1 that appears in $L_\Gamma$ the value $NE(\Gamma(A_1, A_2))$ is learned correctly by Algorithm 2. Therefore the learned layer selection game coincides (up to relabeling) with the true one, and applying Find-CNE to the learned payoff matrix returns the same best response set for Player 1 as in the true game. Hence the CNE strategy for Player 1 is correctly identified.

## C    DISCUSSION OF CAUSAL GAMES & INFORMATION SOURCES

While the Causal Prisoner's Dilemma highlights the difficulty of cooperation – and shows that counterfactual reasoning can improve upon standard Nash-like outcomes – other strands of the literature explore strategic interactions from orthogonal perspectives, including frameworks such as Correlated Equilibrium and Bayesian Games.

The concept of Correlated Equilibrium (CE), introduced by Aumann (1974), generalizes Nash Equilibrium by allowing players to coordinate their strategies through signals from an external correlation device. Unlike in Nash equilibria, where each player optimizes independently, correlated equilibria permit coordinated play, which can yield higher social welfare. This framework is particularly effective in settings where cooperation can be facilitated by signals or mediators without direct communication. In addition, CE is often easier to compute and achieves better efficiency in certain games, especially when compared to independently derived strategies. Applications include traffic routing, bargaining, multi-agent learning, and regret minimization.

Another important extension is Bayesian Games, introduced by Harsanyi (1967), which address strategic interactions under incomplete information. Here, players possess private information about their own types (e.g., preferences, available actions, or payoffs) but maintain beliefs about others' types, often represented by probability distributions. This framework allows players to form and update strategies based on their beliefs. Bayesian Games naturally model scenarios involving uncertainty about the environment or about other agents, such as auctions, signaling games, contract theory, and mechanism design. From a computational standpoint, they introduce additional complexity due to the structure of beliefs and type spaces.

Comparing these frameworks to the standard Nash equilibrium setting shows how they enrich strategic analysis by incorporating coordination (in CE) and information asymmetry (in Bayesian Games). In this section, we offer a preliminary discussion on how these concepts relate to causal reasoning, and how counterfactual thinking may further enhance strategic decision-making in complex environments.

Specifically, we argue that the notion of information, as traditionally understood in the literature, is orthogonal to the causal structure captured in Causal Normal Form Games (CNFGs). This means that the causal framework can be naturally extended to incorporate sources of information available to agents. We first will revisit the standard definitions of normal form games (Sec. C.1), correlated equilibrium Sec. C.2, and Bayesian games (Sec. C.3) using formal causal language. We then introduce CNFGs with information and compare them to these classical game-theoretic frameworks (Secs. C.4 and C.5), highlighting their expressive differences and extensions.

### C.1    STANDARD VERSUS CAUSAL NORMAL-FORM GAMES

To ground the discussion and establish a common denonimator, we start with the definition discussed earlier, first of a causal normal form game (Def. 2.10):

**Definition C.1** (Causal Normal Form Game). A tuple $\Gamma = \langle \mathbb{M}, \mathcal{A}, \mathcal{R} \rangle$ is a Causal Normal Form Game (CNFG, for short), where

- $\mathbb{M}$ is a CMAS $\langle M, N, \mathbf{X}, \mathbf{Y} \rangle$,

- $\mathcal{A} = (\mathcal{A}_1, \ldots, \mathcal{A}_n)$ is the set of policies for the $n$ agents, where $\mathcal{A}_i \in \{\mathcal{A}^1, \mathcal{A}^2, \mathcal{A}^1 \cup \mathcal{A}^2, \mathcal{A}^3\}$,

- $\mathcal{R} = (\mathcal{R}_1, \ldots, \mathcal{R}_n)$ is the set of reward functions.

Next, we provide the standard definition of a normal form game.

**Definition C.2.** A Normal-Form Game is defined as a 3-tuple $G = (N, A, u)$ where

- $N$: set of players.

- $A = A_1 \times \cdots \times A_n$: action space, where $A_i$ is the set of actions available to player $i$.

- $u = (u_1, \ldots, u_n)$: utility functions, where $u_i : A \to \mathbb{R}$.

| Player 2 / Player 1 | $X_2 = B$ | $X_2 = F$ |
|---|---|---|
| $X_1 = B$ | $2, 1$ | $0, 0$ |
| $X_1 = F$ | $0, 0$ | $1, 2$ |

Figure 10: Payoff matrix for Battle of Sexes

| P2 / P1 | $L_1$ | $X_2 = B$ | $X_2 = F$ |
|---|---|---|---|
| $L_1$ | $1.875, 1.875$ | $0.875, 1.375$ | $1.375, 0.875$ |
| $X_1 = B$ | $1.375, 0.875$ | $2, 1$ | $0, 0$ |
| $X_1 = F$ | $0.875, 1.375$ | $0, 0$ | $1, 2$ |

Figure 11: Payoff matrix for Battle of Sexes with $L_1$ and $L_2$ actions

Whenever the policy space is constrained for the interventional layer ($L_2$), CNFG reduces to an NFG.

**Theorem C.3.** *A CNFG with $L_2$ actions can be converted to an NFG and vice versa with the same action space $\mathcal{A}$.*

*Proof.* The proof is constructive. CNFG with $L_2$ actions can be converted to a normal form game, simply by having the $L_2$ interventions as actions and $\mathbb{E}[\mathcal{R}_i(\mathbf{Y}_i(\mathbf{x}))]$ as the utility. The other way can be done as follows: define an action variable $X_i$ for each of the $n$ agents, where $D_{X_i} = A_i$. Then define $Y_i(x) = u_i(x)$ and $\mathcal{R}$ as a set of identity functions. $\square$

Hence, a definition of a Normal Form Games in the causal terms would be:

**Definition C.4.** A Normal-Form Game is defined as a 3-tuple $\Gamma = \langle \mathbb{M}, \mathcal{A}, \mathcal{R} \rangle$ where

- $\mathbb{M}$ is a CMAS $\langle M, N, \mathbf{X}, \mathbf{Y} \rangle$,

- $\mathcal{A} = (\mathcal{A}_1^2, \ldots, \mathcal{A}_n^2)$ is the set of $L_2$ policies for the $n$ agents,

- $\mathcal{R} = (\mathcal{R}_1, \ldots, \mathcal{R}_n)$ is the set of reward functions.

This gives an intuitive understanding of why and how CNFGs generalize NFGs – a claim made in Thm. 2.11. For concreteness, consider the following example.

**Example C.5** (Prisoner's Dilemma). Consider the classical Prisoner's Dilemma game with payoffs as shown in Fig. 7b. To represent this game as a CNFG, define $X_1$ and $X_2$ as the action variables in a CMAS, and $\mathbf{Y} = Y_1, Y_2$ as the corresponding reward signals. Assume $X_1, X_2$ are binary, where 0 represents cooperation (C) and 1 represents defection (D). Define $Y_1$ and $Y_2$ as deterministic functions of $X_1$ and $X_2$, based on the given payoff table. For completeness, let $X_1 = U_1$ and $X_2 = U_2$, where the prior distribution over exogenous variables is $P(U_1 = 1) = P(U_2 = 1) = 0.5$. Note that this distribution is defined for formal completeness but does not affect the $L_2$ payoffs.

To convert this CNFG into an NFG, we focus on $L_2$ actions. The $L_2$ policy space $\mathcal{A}^2$ maps to the action values of $X_1$ and $X_2$, i.e., either 0 (C) or 1 (D). The payoff for any joint action $(x_1, x_2)$ is given by $\mathbb{E}[\mathbf{Y} \mid do(x_1, x_2)]$, aligning exactly with the utility function in the standard Normal Form representation. $\square$

As discussed earlier, Thm. 2.11 noted that a CNFGs is strictly more expressive than NFGs by showing two CNFGs that agree on the interventional layer but may have different equilibirums, when other layers of PCH are considered.

C.2  CORRELATED EQUILIBRIUM

In this section, we investigate how CNFGs can be extended to systems with information – an important step toward modeling more realistic decision-making scenarios. We begin by examining Correlated Equilibrium through a classical example known as the Battle of the Sexes.

**Example C.6** (Battle of Sexes). A couple of agents want to spend time together in the evening. Agent 1 wants to go to the ballet, while Agent 2 prefers a football match. Their payoffs, based on whether they go to the ballet or football, are shown in Tab. 10. The symmetric Nash equilibrium for this game occurs when both agents go to their preferred location two-thirds of the time, yielding a

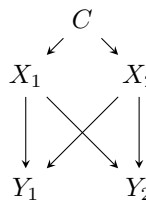

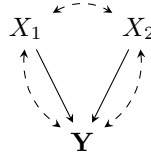

Figure 12: Battle of Sexes with Information from a coin toss

Figure 13: Causal Diagram for Battle of Sexes with unobserved confounding

joint payoff of $(0.75, 0.75)$. We can represent this Normal Form Game as a CNFG with $L_2$ actions. The causal diagram corresponding to the SCM is shown in Fig. 12. The actions of the agent $i$ in this SCM is given by $do(X_i = B)$ and $do(X_i = F)$ for $i \in \{1, 2\}$.

Suppose the players now have access to a coin and observe the outcome of the toss, $H, T$, making decisions accordingly. Assume they follow the strategy $\{H \to B, T \to F\}$ – that is, if the coin shows heads, they go to the ballet; if it shows tails, they go to the football match. Note that this is an equilibrium strategy, as neither player has an incentive to deviate from it. The resulting equilibrium payoff is $(1.5, 1.5)$. Such an equilibrium is called a *correlated equilibrium*, and it is superior to the Nash equilibrium.

We can also represent this graphically in the causal diagram. In Fig. 12, a new variable $C$ can be introduced, which takes two values $\{H, T\}$, each with probability $0.5$. Now, since the outcome of the coin is available to the two agents, they can condition their policies on the outcome of this coin. So, the new policy space of the agents would be a mapping from outcome of coin to the show they want to attend, that is $A_i : \{H, T\} \to \{B, F\}$. If we are talking of mixed strategy, the policy of the agent is given by the distribution $\pi_i(\cdot \mid C)$ over the values $\{F, B\}$. Thus, we can simply represent additions of random variables in a correlated equilibrium as new variables in the causal model (also known as confounders).

We now formally define Correlated Equilibrium using causal language.

**Definition C.7** (Correlated Equilibrium). Given a CNFG $\Gamma = \langle \mathbb{M}, \mathcal{A}^2, \mathcal{R} \rangle$ with policy space restricted to $L_2$, a correlated equilibrium $(\mathbf{S}, P(\mathbf{S}), \pi)$ is a tuple, where $\mathbf{S} = (S_1, \ldots, S_n)$ is a tuple of random variables with distribution $P(\mathbf{S})$ and $\pi = (\pi_1, \ldots, \pi_n)$ is the set of mappings $\pi_i : D_{S_i} \to \mathcal{A}_i^2$ and for each agent $i$ and every other mapping $\pi_i'$,

$$\sum_{\mathbf{s} \in D_{\mathbf{S}}} P(\mathbf{s}) \mathcal{R}_i \big( \mathbf{Y}_{i[X_1 = \pi_1(S_1), \ldots, X_i = \pi_i(S_i), \ldots X_n = \pi_n(S_n)]} \big)$$

$$\geq \sum_{\mathbf{s} \in D_{\mathbf{S}}} P(\mathbf{s}) \mathcal{R}_i \big( \mathbf{Y}_{i[X_1 = \pi_1(S_1), \ldots, X_i = \pi_i'(S_i), \ldots X_n = \pi_n(S_n)]} \big) \tag{27}$$

As observed in the previous example, it is possible to represent the correlated equilibrium as a NE of a CNFG with information. Next, we formally define CNFG where the agents have access to pieces of information (possibly shared) before acting.

**Definition C.8** (CMAS with States). A Causal Multi-Agent System (CMAS) with states is a tuple $\langle M, N, \mathbf{X}, \mathbf{S}, \mathbf{Y} \rangle$, where $M : \langle \mathbf{U}, \mathbf{V}, \mathcal{F}, P \rangle$ is an SCM and

- $N$ is the set of $n$ agents,

- $\mathbf{X} = (\mathbf{X}_1, \ldots, \mathbf{X}_n)$ is the ordered set of action nodes with $\mathbf{X}_i, \mathbf{X}_j \subset \mathbf{V}$ for $i, j \in [n]$ and $\mathbf{X}_i \cap \mathbf{X}_j = \emptyset$ if $i \neq j$,

- $\mathbf{S} = (\mathbf{S}_1, \ldots, \mathbf{S}_n)$ is the ordered set of context nodes $\mathbf{S}_i \subset \mathbf{V}$ for the agent $i$ for $i \in [n]$, and for all $i \in [n]$, $\mathbf{S}_i \notin De(\mathbf{X}_i)$

- $\mathbf{Y} = (\mathbf{Y}_1, \ldots, \mathbf{Y}_n)$ is the ordered set of reward signals, with $\mathbf{Y}_i \subseteq \mathbf{V}$ for all $i \in [n]$. $\quad \square$

Once we have introduced the notion of a CMAS to model the environment, we can consider the information available to the agents about the states.

| Police \ Suspect | $S = 1$ | $S = 0$ |
|---|---|---|
| $P = 1$ | $0, 0$ | $2, -2$ |
| $P = 0$ | $-2, -1$ | $-1, 1$ |

| Police \ Suspect | $S = 1$ | $S = 0$ |
|---|---|---|
| $P = 1$ | $-3, -1$ | $-1, -2$ |
| $P = 0$ | $-2, -1$ | $0, 0$ |

Figure 14: Payoff when suspect is criminal (that is $T = 1$).

Figure 15: Payoff when suspect is civilian (that is $T = 0$).

**Definition C.9** (CNFG with Information). A tuple $\Gamma = \langle \mathbb{M}, \mathcal{A}, \mathcal{R}, \mathcal{I} \rangle$ is a Causal Normal Form Game (CNFG), where

- $\mathbb{M}$ is a CMAS with states $\langle N, M, \mathbf{X}, \mathbf{S}, \mathbf{Y} \rangle$,

- $\mathcal{A} = (\mathcal{A}_1, \ldots, \mathcal{A}_n)$ is the set of policies for the $n$ agents, where $\mathcal{A}_i \in \{\mathcal{A}^1, \mathcal{A}^2, \mathcal{A}^1 \cup \mathcal{A}^2, \mathcal{A}^3\}$,

- $\mathcal{R} = (\mathcal{R}_1, \ldots, \mathcal{R}_n)$ is the set of reward functions.

- $\mathcal{I}$ is the information available to the agents. $\square$

The information $\mathcal{I}$ can take many forms and is introduced to make the definition more complete and general. For example, one form of information might be the distribution over states, $P(\mathbf{S})$, which helps illustrate the relationship between Correlated Equilibrium and equilibrium concepts in CNFGs with states. Other forms of information available to the agents could include interventional or counterfactual distributions. While this is outside the scope of the present paper, it presents a promising direction for future work.

**Theorem C.10.** *If $(\mathbf{S}, P(\mathbf{S}), \pi)$ is a correlated equilibrium of NFG $\Gamma$, then, $\pi$ is a NE of the CNFG with Information designed as follows:*

1. *$\mathbf{X} = (X_1, \ldots, X_n)$ are the actions and $\mathbf{Y} = u(x)$ for each combination are obtained from NFG $\Gamma$. $\mathcal{R}$ is identity for all agents.*

2. *Introduce variables $\mathbf{S} = (S_1, \ldots, S_n)$ with distribution $P(\mathbf{S})$.*

3. *Define the $L_2$ action space $\mathcal{A}_i^2$ for the agent $i$ as soft interventions $\pi(X_i \mid S_i)$*

4. *$P(\mathbf{S})$ is available to the agents and the expected payoff for policies $\pi$ is given by:*

*Proof.* The proof follows from the definition of correlated equilibrium. Since $\pi$ is the policy in the correlated equilibrium, it is also the best response as per Eq. 27. Hence, if every agent is playing the best response given $\mathbf{S}$, we have each agent is playing NE policy (from Def. A.8) in the game with policies conditioned on $\mathbf{S}$. $\square$

C.3  BAYESIAN GAMES

Here, we introduce an additional layer of complexity in the information structure through the concept of Bayesian Games. Before presenting the formal definition, we begin with an example.

**Example C.11** (Sheriff's Dilemma). A police officer faces an armed suspect, and they must simultaneously decide whether to shoot. The suspect could be either a criminal or a civilian, but the officer is unaware of the suspect's true identity. It is preferable for the suspect to shoot if they are a criminal and not to shoot if they are a civilian. However, in hindsight, it is better for the officer to shoot if the suspect shoots – but in reality, they must act simultaneously.

Depending on the type of the player, criminal or civilian, the payoffs corresponding to the actions of the players are shown in Fig. 14 and Fig. 15 respectively. Now, the question is how do we compute the best policy for the agents.

Let's start with the suspect's actions. If the suspect is a criminal, then shooting is a dominant strategy and if the suspect is a civilian, not shooting is the dominant strategy. However, from the perspective

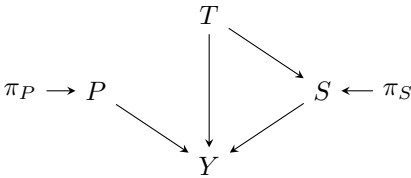

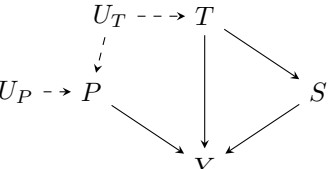

Figure 16: Causal Diagram for Sheriff's Dilemma with $L_2$ policy space.

Figure 17: Causal Diagram for Sheriff's Dilemma with $L_1$ policy space.

of the policmen, things are not so simple, since they do not know the type of the suspect. If they know that the suspect is highly likely to be criminal, then it is better for them to shoot, and otherwise not. Hence, in order to form a decision, they need to have a belief over the likelihood of someone being a criminal.

The scenarios can be represented in a corresponding causal graph shown in Fig. 16. The variable $T$ represents the type of the suspect: $T = 0$ indicates a civilian, and $T = 1$ indicates a criminal. The variable $P$ captures the officer's decision to shoot or not, while $S$ denotes whether the suspect chooses to shoot. If we consider $L_2$ actions only, then the policy space is a soft intervention over the values of $P$. Finally, $Y = (Y_1, Y_2)$ represent the utilities of the officer and the suspect, respectively. The value of $Y$ as a function of $P, T$ and $S$ is shown in Fig. 15 and Fig. 14. In addition, the agent should also have a belief of the likelihood of someone being criminal, that is $P(T)$.

This class of Games where there is an uncertainty about the nature of the game is called Bayesian Game. Formally, Bayesian Games can be defined as follows Harsanyi (1967):

**Definition C.12** (Bayesian Games). A Bayesian Game is a tuple $\langle N, A, \Theta, p, u \rangle$, where

- $N$ is the set of $n$ players indexed by $i$;

- $A = A_1 \times \ldots \times A_n$, where $A_i$ is the action set available to player $i$;

- $\Theta = \Theta_1 \times \ldots \times \Theta_n$ where $\Theta_i$ is the type space for player $i$;

- $p : \Theta \to [0, 1]$ is a common prior over types;

- $u = (u_1, \ldots, u_n)$ where $u_i : A \times \Theta \to \mathbb{R}$ is the utility function for player $i$

Now, we formalize the idea of Bayesian Games in the causal framework.

**Theorem C.13** (Bayesian Games in Causal Framework). *A Bayesian Game is a CNFG with information* $\Gamma = \langle M, \mathcal{A}^2, \mathcal{R}, \mathcal{I} \rangle$, *where $M$ is a CMAS with states.*

*Proof.* The construction follows in the same way as Normal Form Games, but now with introduction of the type variables. The CMAS contains the variables $\mathbf{X}$ corresponding to the actions $A$, state variables $\mathbf{S}$ corresponding to the type $\Theta$, $\mathbf{Y}(x) = u(x)$ and $\mathcal{R}$ is the identity function. The information available to the agents is $P(\mathbf{S})$. $\square$

Now, a best response $\pi_i^*$ to a strategy profile $\pi_{-i}$ in a Bayesian Game is defined as

$$BR_i(\pi_{-i}) = \arg \max_{\pi_i'} EU(\pi_i', \pi_{-i}) \tag{28}$$

**Definition C.14** (Bayes-Nash Equilibrium). A Bayes Nash Equilibrium is a strategy profile $\pi$ such that for all $i$, $\pi_i \in BR(\pi_{-i})$.

### C.4 CNFG AND CORRELATED EQUILIBRIUM

In Sec. C.2, we saw how we can represent correlated equilibrium with $L_2$ actions. However, SCMs can inherently represently two more layers of distribution $L_1$ and $L_3$. Using the more general action space is not only a matter of choice, but can be essential in obtaining a better payoff (that is finding the corresponding equilibrium), as illustrated in the following example.

|  |  | $X_2 = B$ | | $X_2 = F$ | |
|---|---|---|---|---|---|
|  |  | $U_2 = 0$ | $U_2 = 1$ | $U_2 = 0$ | $U_2 = 1$ |
| $X_1 = B$ | $U_1 = 0$ | $3,0$ | $3,3$ | $0,0$ | $0,0$ |
|  | $U_1 = 1$ | $0,0$ | $2,1$ | $0,0$ | $0,0$ |
| $X_1 = F$ | $U_1 = 0$ | $0,0$ | $0,0$ | $0,3$ | $0,0$ |
|  | $U_1 = 1$ | $0,0$ | $0,0$ | $3,3$ | $1,2$ |

Table 2: Battle of Sexes with Unobserved Confounding

**Example C.15** (Causal Battle of Sexes). Considering Ex.C.6, we note that, in reality, the decision to go to the ballet or the football game may be influenced by several external factors. For example, when a new ballet is released, agent 1 – who generally prefers football – may also want to attend the ballet. Conversely, if there is a major football event, such as the Super Bowl, agent 2 may also be happy to join agent 1 for the match. These unobserved factors can therefore influence the preferences of both players. The corresponding causal graph is shown in Fig.13.

In addition, the importance of the event may also affect the payoffs. Let $U_1 = 0$ ($U_2 = 0$) indicate that there is a major football match (ballet performance), and $U_1 = 1$ ($U_2 = 1$) that the football match (ballet) is not particularly significant. Now, both agents' intuitions are given by:

$$X_i = \begin{cases} F & \text{if } U_1 = 0, U_2 = 1 \\ B & \text{if } U_1 = 1, U_2 = 0 \\ F \text{ or } B \text{ with equal probability} & \text{otherwise} \end{cases} \tag{29}$$

This means that if either the ballet or football performance is particularly good, the agents choose to attend that event. If both events are equally good or equally unappealing, they make the decision randomly. The payoff for such an $L_1$ action is $(1.875, 1.875)$. In contrast, if they instead base their decisions on signals from a coin toss, the resulting payoff is $(1.5, 1.5)$ – lower than what they would receive by following their natural intuitions.

We now define a correlated equilibrium over a general CNFG, where the agents' action spaces may correspond to $L_1$, $L_2$, or $L_3$ policies.

**Definition C.16** (Causal Correlated Equilibrium). Given a CNFG $\Gamma = \langle \mathbb{M}, \mathcal{A}, \mathcal{R} \rangle$, a causal correlated equilibrium $(\mathbf{S}, P(\mathbf{S}), \pi)$ is a tuple, where $\mathbf{S} = (S_1, \ldots, S_n)$ is a tuple of random variables with distribution $P(\mathbf{S})$ and $\pi = (\pi_1, \ldots, \pi_n)$ is the set of mappings $\pi_i : D_{S_i} \to \mathcal{A}_i$ and for each agent $i$ and every other mapping $\pi_i'$,

$$\sum_{\mathbf{s} \in D_{\mathbf{S}}} P(\mathbf{s}) \mathcal{R}_i \big( \mathbf{Y}_{i[X_1 = \pi_1(S_1), \ldots, X_i = \pi_i(S_i), \ldots X_n = \pi_n(S_n)]} \big)$$

$$\geq \sum_{\mathbf{s} \in D_{\mathbf{S}}} P(\mathbf{s}) \mathcal{R}_i \big( \mathbf{Y}_{i[X_1 = \pi_1(S_1), \ldots, X_i = \pi_i'(S_i), \ldots X_n = \pi_n(S_n)]} \big) \tag{30}$$

The definition is similar to that of a classical correlated equilibrium, except that the equilibrium is for a CNFG with action spaces that can span across any layer of the PCH.

C.5 CNFG AND BAYESIAN GAMES

Similarly, in Bayesian Games, agents need not be restricted to layer $L_2$, as $L_1$ and $L_3$ policies may potentially yield higher rewards, as illustrated next.

**Example C.17** (Causal Sheriffs Dilemma). Consider Ex. C.11. In reality, the situation may not be so simple or well-defined. Unobserved factors might influence both the officer's assessment of the suspect and the suspect's decision to shoot. For instance, a suspect's background might affect both their likelihood of being a criminal and their behavior. A well-trained officer might intuitively discern such subtle cues and make a quick judgment about whether to shoot. An untrained officer, on the other hand, may lack this ability and be more prone to error. This creates unobserved confounding between the suspect's identity and the officer's tendency to shoot. In other words, the

| Action Space | Payoff depends on agents' actions | Agents act based on a signal | Agents act based on type and has belief about the types |
|---|---|---|---|
| $L_2$ | Normal Form Games (Def. C.2) | Correlated Equilibrium (Def. C.7) | Bayesian Games (Def. C.13) |
| $L_1, L_2, L_3$ | Causal Normal Form Games (Def. 2.10) | Causal Correlated Equilibrium (Def. C.16) | Causal Bayesian Games (Def. C.18) |

Figure 18: Comparison of different representations with information and action spaces

officer may not be able to articulate why they want – or do not want – to shoot, but their instinct carries information about their internal state.

Consider two scenarios, $M_1$ and $M_2$, that induce the same causal diagram shown in Fig. 17. In $M_1$, the officers are well-trained; in $M_2$, they are not. In both scenarios, let an adverse background be denoted by the variable $U_T = 1$, with $P(U_T = 1) = 0.1$. Suppose the suspect is a criminal, that is, $T = 1$ if and only if they come from an adverse background. This background may influence the suspect's behavior, which in turn can influence the officer's decision to shoot. In the causal diagram, this pathways are represented by the dashed arrows.

Further, in scenario $M_1$, the officer is able to pick up on these non-verbal cues, and their probability of shooting is given by $P(P = 1 \mid U_T = 1) = 0.9$ and $P(P = 0 \mid U_T = 0) = 0.9$. In the second scenario, $M_2$, the officer almost always makes mistakes, and their probability of shooting is given by $P(P = U_T) = 0.1$. The payoffs $\mathbf{Y} = (Y_1, Y_2)$, as a function of $P$, $T$, and $S$, are shown in Tables 14 and 15.

Now suppose Congress wants to recommend a new policy by passing a law that determines whether officers should shoot or not. They disregard the officers' natural intuitions entirely and compute the Bayesian Nash Equilibrium (BNE) of the game induced by the model, concluding that it is better if the officer does not shoot at all. The expected payoff for the officer under the BNE is therefore given by:

$$\mu_{\text{BE}} = -2 \cdot 0.1 = -0.2 \tag{31}$$

However, if the law is not implemented, then in $M_1$ and $M_2$, the expected $L_1$-payoff of the policeman are respectively

$$\mu_1 = -0.11, \quad \mu_2 = -0.99 \tag{32}$$

This implies $\mu_2 < \mu_{\text{BE}} < \mu_1$, indicating that, even though both SCMs induce the same Bayesian game, implementing the law would be harmful in $M_1$, while beneficial in $M_2$.

In essence, this is similar to the scenarios illustrated in Ex. 1.1. Now, we can rewrite the definition of Causal Bayesian Games without restricting ourselves to $L_2$ layer.

**Definition C.18** (Causal Bayesian Games). A Bayesian Game is a CNFG with information $\Gamma = \langle M, \mathcal{A}, \mathcal{R}, \mathcal{I} \rangle$, where $M$ is a CMAS with states.

Note that even if two CMASs coincide on their $L_2$ distributions, they may differ in their $L_1$ and $L_3$ distributions – particularly in the corresponding $L_1$ and $L_3$ actions. This is formalized below.

**Theorem C.19.** *Given a Bayesian Game, there exists two Causal Bayesian Games $\Gamma_1$ and $\Gamma_2$ with expected $L_1$ payoffs $\mu_1$ and $\mu_2$ and Bayes-Nash Equilibrium (BE) payoffs $\mu_{BE}$, such that*

$$\mu_2 \leq \mu_{BE} \leq \mu_1 \tag{33}$$

*Proof.* The construction is similar to the one in the proof of Thm. 2.11. $\square$

|  |  | $X_2 = 0$ | | $X_2 = 1$ | |
|---|---|---|---|---|---|
|  |  | $U_2 = 0$ | $U_2 = 1$ | $U_2 = 0$ | $U_2 = 1$ |
| $X_1 = 0$ | $U_1 = 0$ | $-2, 2$ | $-2, -6$ | $-2, -6$ | $-2, 2$ |
|  | $U_1 = 1$ | $2, -2$ | $-4, 0$ | $-4, 0$ | $2, -2$ |
| $X_1 = 1$ | $U_1 = 0$ | $2, -2$ | $-4, 0$ | $-4, 0$ | $2, -2$ |
|  | $U_1 = 1$ | $-2, 2$ | $-2, -6$ | $-2, -6$ | $-2, 2$ |

Table 3: $Y_1, Y_2$ as a function of $U_1, U_2, X_1, X_2$ for SCM in Table 2b

The above discussion illustrates the fact that information structure and actions based on layers of PCH are orthogonal to each other. We can have one without the other and even both of them in the same model, without compromising the other. The summary of the axis can be shown in Fig. 18.

# D    Additional Examples and Discussion

## D.1    SCM for Table 2b

Consider the SCM with $\mathbf{U} = \{U_1, U_2\}, \mathbf{X} = \{X_1, X_2\}$ and $\mathbf{Y} = \{Y_1, Y_2\}$. The domains of $U_1, U_2, X_1$ and $X_2$ are $\{0, 1\}$. $P(U_1 = 0) = P(U_2 = 0) = 0.5$. $X_1 = U_1$ and $X_2 = U_2$. $\mathbf{Y}$ as a function of $U_1, U_2, X_1, X_2$ are shown in Table 3.

The action space available to Player 1 and Player 2 are $\mathcal{A}^3$ and $\mathcal{A}^1 \cup \mathcal{A}^2$ respectively.

## D.2    Assumptions for Alg. 2

For the algorithm to work, we will make the following assumptions. Assume that the learning is from the perspective of Player 1.

**Assumption D.1** (Identifiability of Mixture). Let $\mathbf{Y}_{x_1, x_2} \mid x_1', x_{2i}' \sim \phi_i$, for $i \in k$, where $\phi_i$ is a distribution dependent on $x_1', x_1, x_2$ and $k = |D(X)|$. We assume that the distributions are such that their mean and weights are identifiable from their mixture upto a permutation of the $i$'s:

$$\sum_{i=1}^{k} p_i \phi_i(x_1', x_1, x_2) \tag{34}$$

or, the distributions are same for all $i \in [k]$.

Next, we show some example distributions and conditions that satisfy the above assumption.

**Example D.2** (Deterministic Function). Consider the case when $P(\mathbf{Y}_{x_1, x_2} \mid x_1', x_2')$ has all its mass on a single point. In addition, assume that

$$E[\mathbf{Y}_{x_1, x_2} \mid x_1', x_{2i}'] \neq E[\mathbf{Y}_{x_1, x_2} \mid x_1', x_{2j}']$$

for $i \neq j$. Then, for each $(x_1', x_1, x_2)$ we will get $k$ distinct values of $\mathbf{Y}$, and we can map each $(\mathbf{Y}_i, x_1', x_1, x_2)$ to a particular $i$ and $p_i = P(\mathbf{Y}_i \mid x_1', x_1, x_2)$ for $i \in [k]$.

**Example D.3** (Gaussian Mixtures). Yakowitz & Spragins (1968) showed that mixture of multivariate Gaussians are identifiable. Hence, we can get the mixing proportions and the mean of the Gaussians from the sufficient amount of data.

The next assumption ensures that Player 1 can correctly deduce the intuition of the other player from the observations.

**Assumption D.4.** For all assignments $x_2', x_2''$ to the natural intuition of the second player $P(x_1', x_2') \neq P(x_1', x_2'')$.

Note that if $P(x_1', x_2')$ are sampled from a continuous distribution then the assumption is true almost surely.

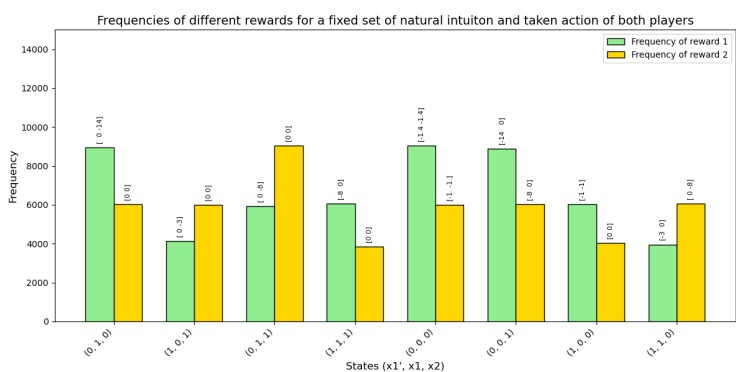

Figure 19: Frequencies of rewards observed for a particular tuple $(x_1', x_1, x_2')$

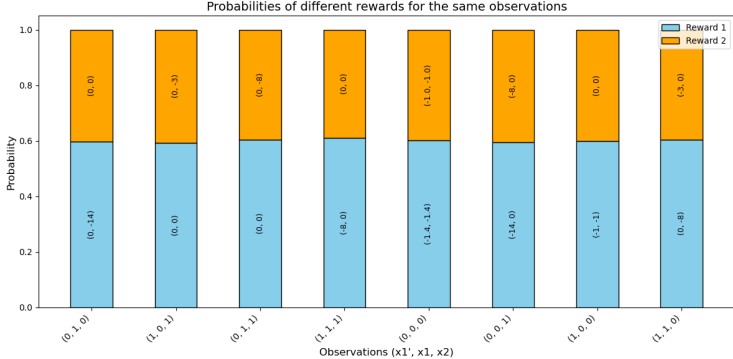

Figure 20: Probabilities of the rewards observed for a particular tuple $(x_1', x_1, x_2')$

Table 4: Payoff Matrix learned by Player 1 in Causal Prisoner's Dilemma

|  | $X_2 = X_2'$ | $do(X_2 = 0)$ | $do(X_2 = 1)$ | $X_2 = 1 - X_2'$ |
|---|---|---|---|---|
| $X_1 = X_1'$ | $-2.443, -2.450$ | $-1.218, -2.684$ | $-8.892, 0.000$ | $-7.668, -0.239$ |
| $do(X_1 = 0)$ | $-2.683, -1.235$ | $-0.983, -0.983$ | $-6.932, -0.490$ | $-5.232, -0.239$ |
| $do(X_1 = 1)$ | $0.000, -8.848$ | $-0.475, -6.951$ | $-1.960, -1.897$ | $-2.435, 0.000$ |
| $X_1 = 1 - X_1'$ | $-0.240, -7.637$ | $-0.240, -5.250$ | $0.000, -2.387$ | $0.000, 0.000$ |

### D.3 CTF-NASH LEARNING ON CAUSAL PRISONER'S DILEMMA

This section shows the results of applying `Ctf-Nash-Learning` on Causal Prisoner's Dilemma. The experiment was carried out on 100K samples of $(x_1', x_1, x_2, \mathbf{y})$ when both agents were playing `Ctf-RCT`. The rewards were assumed to be deterministic, that is, $P(\mathbf{y}_{x_1, x_2} \mid x_1', x_2')$ has a point mass. Now, for each tuple $(x_1', x_1, x_2)$ the frequencies of $\mathbf{y}$ obtained are shown in Fig. 19. For example, when $(x_1', x_1, x_2')$ is $(0, 1, 0)$, then the reward $(0, -14)$ was observed nearly 9000 times while $(0, 0)$ was observed 6000 times, and when it is $(0, 1, 1)$, then the reward $(0, -8)$ occurs nearly 6000 times and $(0, 0)$ occurs nearly 9000 times, and so on.

From this frequency table, we can compute the probabilities as shown in Fig. 20. For instance, when $(x_1', x_1, x_2')$ is $(0, 1, 0)$, the two values of $x_2'$ are taken with a probability of roughly 0.6 and 0.4. The same is observed for the tuple $(0, 1, 1)$. Hence, one value of $x_2'$ occurs with a probability of 0.6 and the other with 0.4. However, there is no way to know whether it is 0 or 1 that occurs with a probability 0.6. This results in a permuation of the actions. For example, if $x_2'$ is correctly identified, that is $\hat{x}_2' = x_2'$ then $(\hat{x}_2' = 0, x_2 = 0), (\hat{x}_2' = 1, x_2 = 1)$ would correspond to the natural instinct or action $(X_2 = X_2')$. However, if they are not correctly identified, that is $\hat{x}_2' = 1 - x_2'$, then the same would correspond to acting against intuition $(X_2 = 1 - X_2)$.

The learned payoff matrix is shown in Table. 4. Even if the actions are permutations of the original payoff matrix, the equilibrium remains same. Hence, the algorithm will be able to find the equilibrium correctly.

The code is available at `https://anonymous.4open.science/r/CGT-2025/`.

### D.4 FORGETTING IN PRISONER'S DILEMMA

As noted in Sec. 3, it is not always in the best interest of the agents to forget. Consider the classical prisoners dilemma: the choices of the action spaces are $\{C\}, \{D\}$ and $\{C, D\}$ and forget about whatever is not included in the sets. The metagame over the choice of the action spaces is shown below.

| P1 \ P2 | $\{C\}$ | $\{D\}$ | $\{C, D\}$ |
|---|---|---|---|
| $\{C\}$ | $-1, -1$ | $-7, -0.5$ | $-7, -0.5$ |
| $\{D\}$ | $-0.5, -7$ | $-1.9, -1.9$ | $-1.9, -1.9$ |
| $\{C, D\}$ | $-0.5, -7$ | $-1.9, -1.9$ | $-1.9, -1.9$ |

Table 5: Extended Prisoner's Dilemma with a third action CD

Note, if both the agents forget about defecting $D$, and plays only $C$, then one of the agents can move to the action space $\{C, D\}$ and get a better payoff while the other is worse of. The resultant NE of this metagame thus also turns out to be $(-1.9, -1.9)$ with action spaces $\{D\}$ or $\{C, D\}$. Thus in classical prisoners dilemma, agents do not have advantage with forgetting.

## E    FAQ

**1. What are $L_1$, $L_2$, and $L_3$ actions, and why are they essential in modeling decision-making?**

**A:** Pearl (2009) and Bareinboim et al. (2022) introduced a framework for studying real-world systems – ranging from experiments in medicine to analyses of climate models – using causal models such as Structural Causal Models (SCMs). An SCM induces three levels of distributions: observational ($L_1$), interventional ($L_2$), and counterfactual ($L_3$). One of the key results in this literature is the *Causal Hierarchy Theorem* (CHT), which states that these three levels of distributions form a strict hierarchy and do not collapse. That is, given an $L_1$ distribution, there may be multiple SCMs that induce the same $L_1$ distribution but yield different $L_2$ distributions. Likewise, given both $L_1$ and $L_2$ distributions, multiple models may still differ in their induced $L_3$ distributions (Bareinboim et al., 2022, Thm. 1).

In decision-making systems, an agent interacts with the environment (and possibly with other agents) through its actions. If the agent does nothing and simply observes, this behavior corresponds to an $L_1$ action. If it disregards its intuition and performs an intervention (either hard or soft) it engages in an $L_2$ action. If the agent's realized action depends on what it would have done under natural circumstances (i.e., its $L_1$ action), then the behavior corresponds to a counterfactual, or $L_3$, action. These distinctions in decision-making have been studied extensively over the past decade, including in Bareinboim et al. (2015; 2024).

A particularly relevant work that forms the basis for our discussion in the single-agent setting is Bareinboim et al. (2024), which introduces a scenario known as the Greedy Casino (reviewed in Appendix A). In this setting, there are machines that may blink and patrons who may be drunk. If a machine is blinking and a patron is intoxicated, the patron is more likely to play that machine instinctively (i.e., subconsciously). Conversely, if the patron is sober and the machine is not blinking, they tend to prefer a different option. This behavior reflects natural predispositions, biases, and inclinations, and is modeled as an $L_1$ action.

Of course, this is a stylized example, but human agents often behave in similar ways – acting without full awareness of the underlying causes of their decisions. This phenomenon has been extensively studied in behavioral decision-making and cognitive psychology, most notably in the work of Daniel Kahneman and collaborators (Kahneman, 2003). In contrast, an $L_2$ action in a single-agent setting may involve flipping a coin and allowing the coin toss to determine which machine to play. This process effectively averages over the agent's internal biases and corresponds to what is formally called the causal effect.

These $L_1$ and $L_2$ behaviors are fundamentally distinct from a counterfactual scenario, in which an agent intends to play machine $X = x$ but ends up playing $X = x'$. Formally, this is represented by the counterfactual quantity $P(Y_{X=x} \mid X = x')$. Note that if the agent naturally plays $X = x'$, this implies they were not initially inclined to play $X = x$ (see Fig. 2a for an illustration). This type of counterfactual evaluation was made possible by the introduction of counterfactual randomization Bareinboim et al. (2015), precisely to decouple the agent from its natural flow and enable the estimation of $Q$. [1] The theoretical limits of which counterfactuals can be physically inferred from the world have been recently characterized Raghavan & Bareinboim (2025b). In positive cases, once the counterfactual randomization step is performed, the agent's intuition becomes a new type of information, which can then be conditioned upon.

**2. Why are Causal Models essential if they can be converted into a matrix game with counterfactual actions?**

**A:** Structural causal models (SCMs) are essential for constructing the payoff matrix (e.g., Fig. 3). In particular, determining the payoffs corresponding to $L_1$ and $L_3$ actions inherently relies on the underlying causal model. Once this matrix is computed, it may be viewed as a normal-form game. However, there are four critical phases where causal modeling plays a vital role:

- **Representation:** Traditional game theory lacks the concept of natural actions; it primarily deals with interventions, corresponding to $L_2$ actions in our causal framework. Once the existence of natural actions is acknowledged, the action space expands to include $L_1$, $L_2$, and $L_3$, enabling the construction of a richer payoff matrix.

  Example 1.1 illustrates this distinction more explicitly: two SCMs may yield identical payoffs and equilibria in the $L_2$ action space, yet diverge significantly when $L_1$ and $L_3$ actions

---

[1] There are intriguing connections here to the notion of free will (see Pearl's 2013 discussion, "The Curse of Free Will and the Paradox of Inevitable Regret").

are considered. This expanded matrix – and its associated equilibria – is difficult to recover without causal assumptions or an underlying structural model. For instance, suppose we observe repeated instances of the scenario in Ex. 3.1 and attempt to infer the payoffs for the actions $(C, C)$, $(C, D)$, $(D, C)$, and $(D, D)$ from observational data. We might obtain $(-1.4, -1.4)$, $(-8, 0)$, $(0, -8)$, and $(0, 0)$, respectively. However, if agents follow a randomized controlled trial (RCT) protocol, the corresponding payoffs could be $(-1, -1)$, $(-7, -0.5)$, $(-0.5, -7)$, and $(-1.9, -1.9)$. An underlying causal model with unobserved confounders can explain this discrepancy. The three layers of the causal hierarchy – observational, interventional, and counterfactual – were formally introduced in Bareinboim et al. (2022).

- **Agency and Execution:** Causal modeling also addresses the practical question of how agents can implement these actions. While the payoff matrix in Fig. 3.1 encodes the outcomes, it does not specify the mechanisms by which those actions are executed. From prior results in causal decision theory, we know that $L_1$ actions correspond to passively observing the system, $L_2$ actions to standard interventions (e.g., RCTs), and $L_3$ actions require more advanced techniques, such as counterfactual randomization (e.g., ctf-RCT). Thus, causal modeling provides a bridge between abstract game-theoretic strategies and their realizability in practice. (In fact, the notion of counterfactual realizability – achieved through a specificd type of randomization – appears to exhaust what is physically implementable in the real world; for a more refined discussion, see Raghavan & Bareinboim (2025a).)

- **Solution Concept:** Our solution concept goes beyond merely computing a Nash equilibrium over an expanded action space. Causal modeling introduces a hierarchy of action spaces, allowing agents to commit to specific layers (e.g., $L_1$, $L_2$, $L_1 + L_2$, $L_3$) and ignore the others. This gives rise to a new "metagame," where the strategic choice involves selecting a layer of the PCH, and the equilibrium is computed within the corresponding subspace. This layered structure adds a new dimension to strategic reasoning and highlights the importance of causal structure in shaping the space of strategic possibilities.

- **Learning:** Normal-form representations overlook the structure linking agents' intuitions to their executed actions, whereas CNFGs can capture this relationship directly. In practice, agents may not observe the other agent's intuition when learning the payoff matrix. In such cases, this structural distinction becomes essential, and is explicitly exploited in Alg. 2.

### 3. Is the causal graph necessary? How does it help?

**A:** Causal graphs contain information about the structural dependencies between variables. For example, when the SCM is Markovian in a CNFG, the optimal actions or equilibrium strategies will always lie in $L_2$. However, when Markovianity does not hold, the optimal strategies may fall anywhere between $L_1$ and $L_3$. Without further knowledge of the parameters – either through prior knowledge or interaction with the system—it is impossible to determine which strategy is better.

### 4. How does this work relate to prior works?

**A:** Our goal in this paper is to develop a model that captures both instinctive and deliberate decision-making processes of the human mind. Structural Causal Models provide a principled framework for representing reality as closely as possible. Some prior works have attempted to reconcile causal models and game theory. However, due to differing goals, their models can be extremely restrictive, as discussed in detail in Appendix A.7.

## F   USE OF LLMs

LLMs were used to polish the writing and to select precise wording in certain places.