# OpenReview forum: "Counterfactual Rationality: A Causal Approach to Game Theory"
_ICLR.cc/2026/Conference — ICLR 2026 Conference Withdrawn Submission_

### Official Review · Reviewer_tcmL · 2025-10-27

**Soundness:** 3
**Presentation:** 2
**Contribution:** 2
**Rating:** 2
**Confidence:** 4

**Summary:**

This is a paper in game theory that carries forward a research program already established in several papers on integrating causal structure within game theory.   The idea is that sometimes exogenous causal factors can lead us to take a decision that is at odds with what rationality and standard game theory would have us do.  The authors represent such exogenous factors using Pearl's theory of causality.  The theory provides levels of reflection that are not present in classical game theory and that may lead players out of dilemmas like the prisoner's dilemma.

**Strengths:**

The paper is clear and the ideas are developed with rigor.   It develops a line of research already present in high level AI conferences.

**Weaknesses:**

This is really a game theory paper.  It's an interesting idea to integrate irrational decisions and rational ones and to provide convincing examples showing that sometimes our irrational decisions give a better payoff.  But the link to representational concerns in AI is very thin.

Much more importantly, I don't get the basic intuition on how irrationality is represented in this framework and even why causality in a Pearl framework is necessary or sufficient.  Thus, I see a potentially important conceptual flaw in the design of the paper.

There are all sorts of unknown factors that may cause our best laid plans to fail, but why do we need to put these exogenous causal factors into games?    To be precise. in the discussion of the prisoner's dilemma (PD), how is M1 capturing an "irrational"
situation?  It's just a different epistemic state about the prisoner's situation from M2, and as such will affect their calculations.  If both are in R1, they will given the belief that they are in a favorable position, they will assign a expected utility to cooperation.  But what I've just described is a belief function that's different from what's standardly assumed in a PD; and thus we really have a different game from a standard PD.   It's something more like Rousseau's stag hunt, in which cooperation may be a rational strategy given a certain epistemic and utility profile.

The authors say that Ui and Ri are unknown to the players.  But if they have no beliefs about R and U, then they can't enter into the decision process.  If they are causes of beliefs Ai and Bi then those beliefs Ai and Bi become part of the calculation.

Really what seems to be in operation here is rather a game with different player types as in epistemic game theory (for a basic reference see Fudenberg and Levine).  Epistemic game theory investigates how assumptions about the players’ beliefs and rationality influence their choices in strategic situations.  We have a player type for 1 in R1 and U1, a type in R1 and U0, and so on.   We then see that there are Nash equilibria for the various types that correspond roughly to the set ups in the figure 1.  The epistemic picture makes this clear   the intuition concerning different epistemic backgrounds of the players whether they are "intuitive" players or level 1 strategic players (standard PD) or level 2 strategic players (have reflected on level 1 type players).  What's important for the example is not causation but rather epistemic profiles of players.  This makes sense, as rationality has to do primarily with internal standards on decision.

The levels of reflection in Section 3.1 are interesting but they also seem to be quite similar to levels of reflection concerning best responses to player types that philosophers and economists have used to refine equilibria as in iterative best response strategies (Michael Franke, Signal to Act dissertation) or in epistemic responses to the PD.   Brown, George W. "Iterative solution of games by fictitious play." Act. Anal. Prod Allocation 13.1 (1951).   Bicchieri, Cristina, 1988a, “Strategic Behavior and Counterfactuals”, Synthese, 76(1): 135–169.

There are also other strands of the literature that might be relevant to this enterprise.  For example, see
Joyce, James, 1999, The Foundations of Causal Decision Theory, Cambridge University Press.

**Questions:**

What happens here to the standard assumption of common knowledge of rationality?

Please comment on how you see causation as crucial in this enterprise, especially in light of causal decision theory work and work on epistemic game theory by people like Battigali and many others.

---

### Official Review · Reviewer_fLAc · 2025-10-29

**Soundness:** 3
**Presentation:** 3
**Contribution:** 2
**Rating:** 4
**Confidence:** 4

**Summary:**

This paper introduces Causal Game Theory (CGT), a framework that integrates causal inference with game theory to model both rational and irrational decision-making. The authors define Causal Normal Form Games (CNFGs) and propose the concept of counterfactual rationality, extending Nash equilibrium through Pearl’s causal hierarchy. They prove the existence and dominance of a new equilibrium notion—Causal Nash Equilibrium (CNE)—and develop algorithms to compute or learn it under full or partial observability. Experiments on a causal version of the Prisoner’s Dilemma show that counterfactual reasoning can outperform standard rational strategies.

**Strengths:**

This paper aims to explore an important issue—the tension between rational and irrational decision-making in multi-agent games, which is often overlooked in traditional game theory. The authors take a causal inference perspective and propose the notion of counterfactual rationality, unifying intuitive and rational behaviors within a formal framework. This approach is novel and insightful, extending the theoretical scope of game theory and offering new analytical tools for multi-agent systems, behavioral economics, and causal reinforcement learning.

**Weaknesses:**

## Writing suggestions

The overall structure of the paper is not very intuitive. It is recommended that the authors add a “Background” section to the main text, separating the literature review currently dispersed across Sections 2 and 3. This would make the research motivation and contributions clearer.

In addition, some sections (e.g., § 3.6) merely restate basic game-theoretic concepts without being referenced later; such content could be moved to the appendix. As a theoretical extension, the paper should also better connect its new framework with more classical game-theoretic notions. For example, introducing elements of belief and information sets from incomplete-information games and explaining how these relate to the proposed L₁, L₂, L₃ levels would strengthen the theory, whether placed in the main text or appendix.

Moreover, the abbreviation “CNE” for Causal Nash Equilibrium is problematic, as it is widely used for Correlated Nash Equilibrium, which may cause confusion in future citations. A different acronym is advised.

---

## Content suggestions

From a substantive perspective, the modeling and experiments are rather limited. The framework is only tested on the Prisoner’s Dilemma using Independent Q-Learning, both of which are very basic settings in learning theory. Since the Prisoner’s Dilemma admits a pure Nash equilibrium solvable by IQL, this setup does not sufficiently demonstrate the theoretical advantages of the new framework. Showing better performance on this single game is also unconvincing, as repeated-game formulations could achieve similar improvements.

The authors are encouraged to evaluate the model on more challenging games, such as mixed-equilibrium settings (e.g., Rock–Paper–Scissors) or incomplete-information extensive-form games (e.g., Kuhn Poker, Leduc Poker). Such environments would better reveal the framework’s effectiveness and scalability. It would also be useful to incorporate classical learning mechanisms such as Fictitious Play (FP) to test compatibility and robustness.

Furthermore, the paper should better illustrate the practical relevance of the algorithm in complex scenarios. For instance, traditional Nash equilibria yield zero expected gain in Rock–Paper–Scissors, whereas the proposed causal modeling might potentially overcome this limitation. Demonstrating improvements on more complex tasks (at least at the Kuhn Poker or Leduc Poker level) would substantially strengthen the paper’s credibility and contribution.

Overall, the current theoretical and experimental settings remain too basic, and the results achieved by these examples overlap with existing work, making it difficult to fully assess the framework’s originality and practical significance.

**Questions:**

Refer to the previous section

---

### Official Review · Reviewer_y6cG · 2025-10-31

**Soundness:** 3
**Presentation:** 1
**Contribution:** 2
**Rating:** 2
**Confidence:** 3

**Summary:**

The paper studies an extension of the standard game theoretic framework to combine rational and  irrational behavior. They formalize their model and show it strictly generalizes standard normal-form games. Finally, they discuss algorithms for learning equilibria in this setting.

**Strengths:**

I find the setting interesting. Real-world agents typically don’t behave rationally and modeling this behavior may be relevant to practical applications.  It is not clear though if this work is primarily around explaining irrational behavior in a descriptive fashion or present prescriptive reasoning (which would then offer an alternative notion of rationality altogether).

**Weaknesses:**

First, I find the work quite tangential to the stated focus of ICLR.  It is either offering an alternative descriptive account of decision-making or a prescriptive one, neither of which ultimately relates to "learning representations".  Seems better suited for AAAI, IJCAI, AAMAS, or EC.

Considering the authors are defining a new model, the text makes little effort towards motivating it.  The notation and formalism feel very dense and typically not well described informally. The paper illustrates all the concepts on the extension of the classical Prisoner’s dilemma, but doesn’t describe the extension well. For example, I am not sure what the L3 action set would represent.

Again for defining a new model, one might expect there to be extensive discussion on related concepts.  For example, how does this relate to more standard game theory concepts such as cognitive hierarchy aka. level-k reasoning?  Or even external and internal regret notions?

In the cognitive hierarchy, level-0 reasoning could be thought of as "intuition" (i.e., A1).  Level-$\infty$ captures the standard NE game theoretic reasoning (i.e., A2).  When level-k reasoning is used MASs, agents often can choose the level they reason at, which is related to the meta-game you describe.

In regret notions, one could think of instinct as the actual policy being followed (whatever the agent is doing, i.e. A1).  External regret is measured by how much better might the agent have done if they had played a fixed action instead of the policy they followed (i.e., A2).  Internal regret measures regret for not swapping one action for another in their policy (i.e., A3).  Having no internal regret implies no external regret as well, and corresponds to correlated equilibria concepts.

Lastly, Theorem 2.11 seems to demonstrate that this direction can't tell us anything.  If the players don't know the underlying SCM (weird that it doesn't even know how it's own actions are caused) then they can't distinguish between ${\cal C}_1$ and ${\cal C}_2$ and yet whether they should play the NE of the NFG depends critically on that unknowable choice!  If one is deliberating whether to follow their "instinct", in what way would following their instinct be instinctual anymore?  I think I am having a lot of difficulty in the "suitcase word" (see Minsky) that is "instinct".  I think it is doing a lot of heavy lifting trying to add meaning where it otherwise wouldn't be.

There are numerous other difficulties in the presentation:
* PCH not defined in the main body.
* In the PD example, how is f_X unknown to the prisoners?  The decision made by the prisoner is their decision but they don't know how they make it?
* "If both prisoners ignore their intuition"  Where does intuition appear in this example?  Is that f_X?  Or is that the variable R and U which doesn't play a role in the payoff?
* "{\bf Pa}(V)" => "{\bf PA_V}" I think?
* "[n]" not defined.
* I suspect that 10 lines is not enough on SCMs for most of the ICLR audience to be prepared to follow the rest of the paper.
* So a CMAS specifies behavior as well as payoff, i.e., the endogenous variables are specified by the SCM and contain the player's actions?
* Where does the "natural function" in Ex 2.4 come from?
* Why make P(U_i = 0) = 0.6?  This just makes it harder to make sense of the example!

**Questions:**

1) Since CNFG generalizes NFG, isn’t Thm 2.11 trivial if we consider two NFGs with all payoffs increased / decreased by some constant?  Shouldn't the CNFG ${\cal C}_1$ and ${\cal C}_2 have to relate to the NFG in some way?

2) What does your framework bring over say extensive-form games with bounded rationality?  Or cognitive hierarchy concepts?  Or various forms of regret (e.g., external, internal, or more recent notions of deviation regret in extensive form games)?

---

### Note · Authors · 2025-12-04

I have read and agree with the venue's withdrawal policy on behalf of myself and my co-authors.